# Exploring Non-Soluble Particles through Innovative Confocal Laser and Scanning Electron Microscopy Techniques

Anthony C. Bernal Ayala[1], Angela K. Rowe[1], Lucia E. Arena[2,3], William O. Nachlas[4], and María L. Asar[2,5]

[1]Department of Atmospheric and Oceanic Sciences, University of Wisconsin, Madison, WI, USA
[2]Facultad de Matemáticas, Astronomía, Física y Computación, Universidad Nacional de Córdoba, Córdoba, Argentina
[3]Observatorio Hidrometereológico de Córdoba, Córdoba, Argentina
[4]Department of Geoscience, University of Wisconsin, Madison, WI, USA
[5]Facultad de Ciencias Exactas, Físicas y Naturales, Universidad Nacional de Córdoba, Córdoba, Argentina

**Correspondence:** Anthony C. Bernal Ayala (crespo3@wisc.edu)

**Abstract.** This paper introduces an innovative microscopy analysis methodology to preserve in situ non-soluble particles within hailstones using a protective porous plastic coating, overcoming previous limitations related to melting the hailstone sample. The method is composed of two techniques: trapping non-soluble particles beneath a plastic coat by using the adapted sublimation technique and then analyzing the particles individually with both Confocal Laser Scanning Microscopy (CLSM) and Scanning Electron Microscopy with Energy-Dispersive Spectroscopy (SEM-EDS). CLSM provides insights into physical attributes like particle size and surface topography, enhancing understanding of ice nucleation. SEM-EDS complements CLSM by offering detailed information on individual particle elemental chemistry, enabling classification based on composition. Strategies to reduce background noise from glass substrates during EDS spectral analysis are proposed. By combining powerful, high resolution microscopy techniques, this methodology provides valuable data on hailstone composition and properties. This information can give insights into hail developmental processes by enhancing our understanding of the role of atmospheric particles.

## 1 Introduction

The analysis of hailstone samples has been a subject of increasing interest owing to its potential to provide valuable insights into microphysics and the development of hailstorms (Jeong et al., 2020; Soderholm and Kumjian, 2023; Wang et al., 2023). Consider a hailstone a miniature chronicle of time, preserving a rich history of its formation and growth within its crystalline structure. Understanding hailstones' microphysics and developmental processes is crucial for improving weather forecasting, mitigation strategies, and climate change modeling (Cintineo et al., 2020; Malečić et al., 2022; Wu et al., 2022; Jiang et al., 2023). Atmospheric particles play a crucial role in hailstone microphysics. Particles may be ingested into convective cloud updrafts, serving as cloud condensation nuclei (CCN) or ice nucleating particles (INP), being necessary to form hydrometeors like cloud droplets and ice crystals that are essential for hail formation. Furthermore, hailstones can accumulate particles from the surrounding environment in the cloud during their growth, which could have sizes larger than those that serve as CCN

and INP and would provide information about the source environments involved in hail formation. Particle size and chemical composition are essential to quantify because they influence the nucleation efficiency and growth rate within hailstones. Characterizing these particles allows us to reconstruct their potential influence on hailstone properties, such as fall speed rates and growth regimes, furthering our understanding of hail formation and its complex interplay with atmospheric conditions.

Previous studies have investigated hailstone composition and properties using various techniques to determine their chemical constituents. For example, Šantl-Temkiv et al. (2013) analyzed hailstones from a thunderstorm in Slovenia, focusing on determining dissolved organic carbon and total dissolved nitrogen in hailstones with a total nitrogen module. They also characterized dissolved organic matter through ultrahigh resolution mass spectrometry, looking into the molecular complexity of hailstones. Overall, they noted a strong link to the local environment, particularly the hailstones' bacterial content, emphasizing a soil-driven origin for aerosol particles. More recently, Kozjek et al. (2023) expanded the analysis of hail in Slovenia by identifying fibrous and abiotic particles within melted hailstones.

Li et al. (2018) similarly examined constituents in melted hailstones. In their study, concentrations of ten ions and six environmental pollutants were compared with the aerosol optical depth retrieved from the Moderate Resolution Imaging Spectroradiometer for 15 hailstorm events in China. They found that particulate matter with an aerodynamic diameter of less than $10\,\mu m$ ($PM_{10}$) was the most likely source of the water-soluble ions, although with considerable variability between hailstones. These $PM_{10}$ particles are usually condensed below the cloud base and lifted higher through the updraft, highlighting a link to a terrestrial source. The concentration of water-soluble ions, metals, and metalloids in hailstones was investigated for the first time in South America by Beal et al. (2021), who analyzed hailstone samples from the triple border of Parana, Brazil, and Argentina using ion chromatography and inductively coupled plasma mass spectrometry. That study's cleaning procedure efficiently removed potential chemical contaminants from hailstones by following procedures defined by Christner et al. (2005). They found that $Ca^{2+}$ was the most abundant ion in their samples and interpreted it to be related to agricultural activities of non-local regions through long-distance transport. However, in their work, as in the aforementioned studies, these approaches rely on the melting of the hailstones, which obscures the spatial distribution of non-soluble particles within the hailstone.

Michaud et al. (2014) took an approach that constrained their analysis to the hailstone's embryo, focusing on three hail-producing storms in the Rocky Mountains of the U.S. They utilized stable isotope analysis to estimate in situ temperatures during freezing and examined particle elemental compositions collected from hailstone embryos through scanning electron microscopy. Similar to Beal et al. (2021), their study followed hailstone cleaning procedures defined by Christner et al. (2005), scrapping the outermost layer with a sterile razor blade rinse with 95% ethanol and then melting to be filtered onto a $13\,mm$ diameter, $0.2\,\mu m$ polytetrafluorethylene membrane filter for elemental analysis. Their study focused on melting and filtering hailstone particles larger than $0.2\,\mu m$ because of their interest in studying ice nucleation by immersion. They found that their hailstone embryos contained biological INP that can freeze water at relatively warm subzero temperatures, suggesting that biological particles have the potential to serve as nucleation sites for hailstone formation.

In the discussed previous literature, hailstones were melted to study soluble compounds or particles, and in the case of Michaud et al. (2014), they used filters to separate the particles. A new method, derived from particle separation by adapted sublimation Arena (2023), is described here to remove the need to melt the hailstone for microscopy analysis while retaining

the particle's position within the hailstone. This new method, therefore, allows the analysis of non-soluble particles within the hailstone sample, preserving all biological and non-biological particles without maximum size restriction to provide a representative particle distribution in the hailstone. This approach allows for microscopy techniques to be employed, as detailed below, enabling the characterization of non-soluble particles physically (particle size and surface topography) and chemically (inferred from elements within the individual particles). This information can be used to improve our understanding of the distribution of particles within hailstones as a factor that influences the growth of cloud droplets' initial stages of ice nucleation and subsequent hail growth.

## 2 Principles of Scanning Microscopy Techniques as Applied to Particle Analysis

### 2.1 Confocal Laser Scanning Microscope (CLSM)

Confocal microscopy is a high resolution optical imaging technique that uses a diffraction-limited spot to produce a point source of light and reject out-of-focus light, allowing for imaging of deep tissues and 3D reconstructions of imaged samples (Mercer, 2005; Turner et al., 2001; Liu et al., 2011; Anderson, 2014; Wang and Larina, 2017; Elliott, 2020; Claxton et al., 2006). A modern confocal microscope includes pinholes, objective lenses, low-noise detectors, fast-scanning mirrors, filters for wavelength selection, and laser illumination (Elliott, 2020). Confocal microscopy has been extensively utilized in various disciplines, including supramolecular chemistry (e.g., Kubota et al., 2020), biology (e.g., Claxton et al., 2006), and materials science (e.g., Schnell et al., 2019) due to its high resolution imaging capabilities. In particular, it is extensively used for studying the intricate structures and functions of biological specimens (O'Connor, 1996; Kaye et al., 2015; Fu et al., 2021). This technique offers several advantages over conventional widefield optical microscopy, including the ability to control depth of field, elimination or reduction of background signal away from the focal plane, and the capacity to collect serial optical sections from thick specimens (Mercer, 2005; Turner et al., 2001; Liu et al., 2011; Anderson, 2014; Wang and Larina, 2017; Elliott, 2020; Claxton et al., 2006). CLSM is, therefore, a powerful tool for analyzing non-soluble particles due to its capacity to acquire three-dimensional images to determine particle size distribution and particle surface information.

### 2.1.1 Scanning Electron Microscopy (SEM) and Energy Dispersive X-ray Spectroscopy (EDS)

Scanning Electron Microscopy (SEM) and Energy Dispersive X-ray Spectroscopy (EDS) are techniques often used together to analyze the surface of a sample. Both techniques and principles are based on the interaction of electrons with different types of matter. SEM works by scanning a focused beam of electrons onto a sample, inducing the emission of secondary electrons and backscattered electrons. These electrons can be detected and used to create a high resolution image of a surface. SEM has extensive applications across various scientific disciplines, such as material sciences (e.g., Zhang et al., 2020), geology/planetary science (e.g., Gu et al., 2020), and biology (e.g., Koga et al., 2021). Renowned for its capability to provide high-resolution imaging (e.g., Figure 1) at scales as fine as 50 $nm$ (e.g., Vander Wood, 2017), SEM has proven invaluable in diverse fields, including hailstone analysis (Michaud et al., 2014). Its imaging capabilities make SEM particularly well-suited for investigating

the morphology and texture of solid materials, allowing for a detailed analysis of surface features. EDS, on the other hand, is a technique that allows the analysis of the elemental composition of a sample. It works by detecting characteristic X-rays emitted from the sample when bombarded with a beam of electrons. Each element has a unique set of peaks on its electromagnetic emission spectrum, which allows the identification of the chemical elements present in the sample (e.g., Reed, 1996; Raja and Barron, 2016). Both techniques have been widely used in materials science, particle analysis (e.g., ice-nucleating, in-snow, and dust particles), and other fields to study the surface of samples and analyze their chemical composition (e.g., Krueger et al., 2004; Bern et al., 2009; Ketterer, 2010; Nam and Lee, 2013; Farber, 2019; Raval et al., 2019; Wagner et al., 2019; Brostrøm et al., 2020; Sanchez-Marroquin et al., 2020; Rausch et al., 2022).

## 2.2 Protected Particle Deposit in Hailstone Samples for CLSM/SEM-EDS Analysis

CLSM and SEM-EDS microscopes were not designed for analyzing hailstone composition in sub-zero conditions. While it is common to find cryo stages on SEMs that facilitate sub-zero investigations and heating/cooling stages on optical scopes for handling frozen samples, issues especially arise when trying to preserve the location of particles within the stone. A specific, relevant example of such limitations is that the vacuum in the SEM chamber used to remove air to prevent electron scattering and produce a clear image would result in the sublimation of the hailstone and the movement of particles from their original location within the stone. Prior hail-focused studies (e.g., Mandrioli et al., 1973; Michaud et al., 2014) have avoided these obstacles by melting and filtering the hailstone's liquid water to determine the distribution of viable microorganisms within the hailstone, which limited the analysis to a specific subset of particles. Therefore, a new technique must be applied to hailstones to preserve their in situ particle distribution including non-soluble particles before taking advantage of these powerful microscopy techniques.

Ice exhibits a sublimation point close to its melting point, making it feasible to separate out particles via sublimation. A particularly effective sublimation procedure, as described in Arena (2023), involves placing a thin section of an ice sample on a substrate within a dry environment, allowing the ice to sublimate gradually while ensuring minimal contamination during and after the sublimation process. Over time, the particles remain deposited on the substrate, facilitating their examination through microscopy techniques. By first coating the ice with an ultra thin layer of porous plastic, thus slowing the sublimation process, any potential drag or contamination of the trapped particles while handling the sample is prevented.

This coating technique was pioneered by Schaefer (1941) and subsequently refined by K. Higuchi (see Higuchi, 1958; Higuchi and Muguruma, 1958) for the creation of plastic replicas of ice surfaces. Both researchers applied thin layers of a coating made of polyvinyl formal (FORMVAR) dissolved in ethylene dichloride to the sample surface to facilitate ice sublimation through the formation of small pores in the plastic. Relatively shorter sublimation times within these pores led to the formation of thermal attack imprints (pits), enabling the examination of the sample's texture. This approach eliminates all water molecules while preserving the original spatial distribution of all particles, regardless of their size or composition. Consequently, the particles separated and trapped by the plastic coating can be examined using microscopic techniques such as CLSM or SEM-EDS at room temperature. This coating and sublimation technique serves as the basis for this present study. The following sections describe the novel application and expansion of this approach, from hail collection to microscopy

analysis, toward investigating particle's (now including non-soluble) physical-chemical characteristics and distributions within hailstones.

## 3 Materials and Methods

### 3.1 Hailstone Collection

The hail collection procedure was an effort between the Universidad Nacional de Córdoba (UNC) and Argentinian citizens around the Córdoba Province. When Argentinian citizens collected hail, they took pictures with their hands or some other object as a reference and stored them in their fridges (commonly at -13 $°C$). Argentinian citizens then contacted the Facultad de Matemática, Astronomía, Física y Computación (FAMAF) at UNC or Dr. Lucia E. Arena directly through COSECHEROS APP, phone and WhatsApp to report the hail they collected. Once the hail report was made, a visit was scheduled with the person who collected the hailstone. During the visit, location coordinates, time of collection, home freezer temperature, and information on whether it rained before, during, or after the hail is noted. The name of the person who collected the sample is recorded, and a document is signed that states that the individual is transferring the hail sample ownership to FAMAF for analysis.

The hailstone sample was transferred and kept in a nylon bag in a thermally reinforced cooler where the temperature was controlled at -15 $°C$. This temperature is chosen because the crystallographic structure changes are prolonged (the significant changes in crystallographic structure at this temperature may not start to happen after a year or more). This timeline, however, is not important for the type of analysis described in this paper because it is focused on particle composition, which is not affected by the storage time and not the crystallographic properties of the hailstone. Each sample was designated an alphanumeric identifier that included the name of the person who collected the sample. Once it reaches the laboratory, the hailstone surface is brushed (to remove contamination) and transferred to a cold chamber with a controlled temperature of -15 $°C$. No additional cleaning procedure is considered since the hailstone has not melted, and the risk of further contamination is minimal. This hail collection and conservation procedure became the Argentine provincial citizen science program "Cosecheros de Granizo," which started in October 2018 (Arena and Crespo, 2019; Crespo, 2020).

### 3.2 Hailstone Preparation with an Adapted Sublimation Method

After cleaning subzero facilities (refer to Appendix A), hail samples were processed in the subzero facility Laura Levi Atmospheric Physics Laboratory of FAMAF-UNC, one of the few facilities in the world with the equipment to process hail. Upon the chamber reaching a temperature of -12 +/- 2 $°C$, the chamber should be quickly entered to minimize heat exchange and maintain this required operating temperature for hailstone processing. After entering the chamber, the following hailstone measurements should be made first before the cutting of the hailstone if needed for additional physical analysis of the stone: lengths of the lobes, height and width, symmetry, and weight.

The overall objective of the hailstone preparation process is to expose the embryo by cutting the hailstone. However, due to the coating machine's limited size, smaller thin-section glasses are needed to fit the coating chamber. This strategic preparation ultimately serves the larger goal of entrapping the hailstone particles within a FORMVAR layer, a crucial step for subsequent microscopic analysis. For this reason, a glass substrate is needed to affix the hailstone before cutting it into sections for analysis. Toward this goal, several thin glass sections, assembled in a mosaic pattern (Figure 2), were affixed to a larger piece of glass using ultrapure liquid water. Glass was selected as the substrate for particle deposition due to its transparency to visible light, enabling the examination of particles using transmitted light microscopes (i.e., CLSM). The hailstone was then attached to the glass by applying heat to the bottom of the hailstone to slightly melt the bottom surface, positioning it on the glass, then allowing it to freeze once more (within minutes) until firmly bonded to the glass surface. The glass is too thin to firmly grab for cutting the stone, so the hailstone, now firmly attached to the glass, is further affixed to a thick metal section plate using the same ultrapure liquid water (Figure 3-A). The hailstone sample was then cut over the equatorial symmetric plane using a diamond-encrusted cutting disk of 12 $cm$ diameter with a cutting width of 2 $mm$ to expose the embryo (Figure 3-B). While alternatives to the mentioned disk are plausible (different sizes and cutting widths), they must adhere to specific criteria to achieve the best hailstone cutting performance: (1) the blade has to be as thin as possible to minimize removing hailstone mass during cutting, and (2) its edge must be adequately broad to facilitate cutting larger hailstones. After cutting, the sample is evenly polished with a microtome (Figure 4-A) to provide a thin, even ice layer over the embryo (Figure 4-B).

At this stage, pictures were taken to record the location of growth rings relative to the embryo. Hence, we know where particles will be located relative to the embryo, allowing for future crystallographic and ice growth analysis. After taking pictures of the polished hailstone sample, a layer of 1% FORMVAR solution diluted in ethylene dichloride (refer to Appendix B) is applied to the sample's flat polished surface using a clean glass rod. This application is done in two ways: 1) by dipping one side of the rod into the FORMVAR solution and spreading a small amount over the surface, or 2) by pouring small amounts of the solution onto the surface and evenly spreading it across the polished hailstone. Once the entire surface is covered with the FORMVAR solution, the sample is left to curate for a few minutes and then is ready for sublimation. The hailstone is left in a sealed low-humidity container with silica gel (Figure 4-C) at the -12 $°C$ operating temperature to sublimate it. This process enabled the gradual sublimation of ice over a 24- to 48-hour period, with daily sample monitoring, while also capturing non-soluble particles beneath the layer of the FORMVAR solution. During sublimation, dissolved components may undergo precipitation. For instance, in cases where the original hailstone possessed a brine-like composition, the sublimation process could result in the precipitation of once-soluble salt particles. However, in convective systems, the behavior of soluble components such as sodium chloride and other crystalline inorganic salts is influenced by multiple factors, such as solution properties and convective transport processes (e.g., updraft speed). When these get injected into an updraft, they have the potential to remain undissolved, acting as INPs under certain conditions (Patnaude et al., 2021). Consequently, discerning between particles that were initially non-soluble and those that became non-soluble post-sublimation poses a challenge and is not the primary focus of this paper. However, it is known to a certain degree that some particles were not originally precipitated from the sublimation process owing to their specific composition determined by the SEM-EDS technique described in the next section.

### 3.3 Microscopy analysis

#### 3.3.1 CLSM analysis

Once the sample is completely sublimated and particles are trapped in the FORMVAR, it is ready for microscopy analysis. An OLYMPUS LEXT OLS4000 Confocal Laser Scanning Microscope (CLSM) is used to create a 2-D cross section of the sublimated hailstone (centered on the embryo location) along an axis in the equatorial plane using the lowest available magnification (108x) to locate the non-soluble particles trapped in the FORMVAR. Within this cross-section (Figure 5-A), subsectors are selected with respect to the embryo center for higher magnification (Figure 5-B), with the color image (Figure 5-C) providing high resolution 3D particle surface topography and, therefore, size (Figure 5-E) of individual particles within all subsectors of the scanned 2-D cross section. The magnification is dependent on the size of the particle to be analyzed. It is estimated that using the modified sublimation technique, particles trapped in FORMVAR can be observed using the CLSM as small as 1 $\mu m$, requiring magnification of 2132x. Each particle is individually labeled for further analysis. CLSM files undergo processing using ProFilm software (e.g., as displayed in Figure 5-E), facilitating the extraction of essential particle information. During this stage, particle size is determined by assessing the maximum length along the x and y axes. The obtained data on particle size, coupled with the spatial distribution of each particle, lays the foundation for the subsequent SEM-EDS analysis, as detailed in the following section.

#### 3.3.2 SEM-EDS analysis

The 2-D cross section created with the CLSM (Figure 5-A) is used as a reference point to investigate the elemental composition of the individual particles. Elemental analysis with SEM-EDS was conducted on electrically conductive samples under high vacuum conditions to enhance imaging quality and the accuracy of chemical measurements. This decision was influenced by the absence of a variable pressure mode (50 $Pa$) in the SEM instrument used in the development of this method. Even if such an instrument were available, a high vacuum environment would still be preferred. SEMs equipped for low vacuum analysis introduce a controlled gas, such as nitrogen, or allow some oxygen from the surrounding air to enter the chamber, potentially leading to aerial contamination. Without a dedicated oxygen source or air filtration system, SEM chambers risk contamination from atmospheric impurities. This is because SEMs do not have built-in air filtration systems. These impurities can settle on the FORMVAR coat. Without proper reference from CLSM or SEM, they could be mistakenly included in the particulate selection analyzed with this method.

Moreover, under low vacuum conditions, beam skirting can occur (Goldstein et al., 2017), wherein the gaseous environment alters the profile of the primary electron beam. This alteration typically divides the electron beam into two fractions: an unscattered beam with the original distribution profile and diameter and a scattered beam forming a "beam skirting" around it. This modification occurs prior to reaching the particle surface, affecting the resolution of high-resolution imaging and spectral analysis through EDS. Consequently, a high vacuum environment is recommended due to concerns regarding contamination, lack of beam trajectory control, and the impact on resolution and spectral analysis.

To produce a fully conductive surface for most accurate X-ray chemical analysis, the glass slides with hailstone particles were coated with 25 nm Au and analyzed at $10^{-4}$ $Pa$ chamber pressure. Gold coating was selected over other common coating materials (Au, Au-Pd Alloy, Pd, Pt, Ir, C, and Ag) to enable characterization of C-bearing non-soluble particles without X-ray interferences at the C Ka position or other elements that may exist in hailstone particles. The sample was analyzed with a ZEISS FEG-SEM Sigma with an EDS X-MAX 80 $mm^2$ detector at the Laboratorio de Análisis de Materiales por Espectrometría de Rayos X (LAMARX) facilities at UNC.

Analysis of micron scale non-soluble particles involves optimizing both the spatial resolution of electron imaging and the activation volume of X-ray analysis. Selecting the accelerating voltage is critical to determining the ultimate resolution of imaging and X-ray analysis. The accelerating voltage of the primary beam determines the wavelength of electrons, and higher voltages are generally advised for enhanced spatial resolution in electron imaging (Goldstein et al., 2017). However, it is important to consider that this principle may not universally apply to all materials. In some cases, excessively high voltages could result in electron expansion or penetration into the materials, potentially diminishing resolution. Therefore, it is essential to determine the optimal voltage, potentially opting for a lower one, when analyzing specific samples to ensure optimal imaging resolution. By balancing the energy of the incident beam, it is possible to excite all X-ray lines of interest from the smallest possible volume. Additionally, depending on the object being analyzed, increasing the voltage increases the risk of beam damage to the particle (e.g., Figure 6), thereby jeopardizing the sample's integrity and subsequently influencing the conclusive outcomes of this method. On the other hand, the working distance can affect the depth of the field and beam diameter, which will also affect the clarity and resolution of the image of an object. Optimizing the accelerating voltage and working distance will define the clarity, detail, and accuracy of the imagery obtained from an SEM.

The working conditions of accelerating voltage of 15 $kV$ and 8.5 $mm$ working distance were set for creating the secondary electron image (also referred to as a micrograph) shown in Figure 7-A. For these conditions, at a magnification of 1.14 $kx$, particles as small as 1 $\mu m$ trapped in the FORMVAR solution are observed in the SEM. These SEM-identified particles are then compared to CLSM image locations to ensure the same particle from the CLSM imagery is the one being targeted in the SEM (e.g., Figures 6-C and 7-A) for EDS spectrum analysis.

EDS analysis is conducted for every particle identified in the SEM imagery to determine major and minor element composition of the non-soluble particles trapped in the hailstone sample (Figure 7-B) using the same voltage defined for electron imaging. The choice of 15 $kV$ ensures that heavier elements are included in the EDS analysis by exciting the K lines of elements such as Fe. However, owing to attaching the hailstone to a glass substrate, a lower voltage may reduce background noise. Specific to soda-lime glass, elements such as Si, Na, Ca, Mg, K, and Al may be introduced into the EDS spectral analysis from the glass. To evaluate this effect, Monte-Carlo Simulations of Electron-Specimen Interactions were conducted for 15 and 8 $kV$ with 0.25 $\mu m$ beam diameter using the CASINO program (v2.5.1.0; Drouin et al., 2007) (Figure 8). Results indicate that for a small grain of biotite particle, with a 7 $nm$ layer of FORMVAR, some X-rays are emitted from the glass substrate for particles smaller than 1.5 $\mu m$ at 15 $kV$ that can be minimized by using the 8 $kV$ accelerating voltage. While this suggests that the soda-lime glass can exert minimal impact on the spectral readings for particles down to 1 $\mu m$ minimum size at 8 $kV$, a single-point analysis approach can further reduce the potential spectral contamination.

An example of the single-point technique is highlighted in Figure 7-A, where the EDS measurement is first taken at a point within the red circle where no glass substrate is present (i.e., focused entirely on a section of the particle). Multiple single-point measurements made at least 10 $\mu m$ apart prevents overlap and captures composition variability through the particle. Multiple EDS point measurements are essential for assuring representative particle composition for particles showing varying optical properties in the CLSM color imagery (e.g., Figure 5-D). Figure 7-B shows the corresponding output from the SEM-EDS analysis of the particle in Figure 7-A. Micrographs of individual particles (e.g., Figure 7-A) and EDS data are crucial in particle classification (e.g., carbonaceous [biological, organic/inorganic], silicates) based on elemental composition. Additionally, the high resolution geometry information of each individual particle that can be compared to universally known standards to increase confidence of the particle classification. Micrographs are also complementary in ensuring that EDS spectra are accurately attributed to the same particle analyzed using the CLSM.

## 4 Application of the Microscopy Technique to Hailstones

The outcomes of the methodology are first applied to a 4 $cm$ hailstone (Figure 9) and are introduced here to gain insights into the physicochemical properties of non-soluble particles. This hailstone originated from a supercell storm in Villa Carlos Paz, Argentina, on 8 February 2018 (Kumjian et al., 2020; Bernal Ayala et al., 2022). Using our method, we uniquely determine non-soluble particles' sizes and elemental composition relative to their location in the hailstone. For this methodology, randomly selected particles with a diameter larger than 1 $\mu m$ were analyzed. As shown in Figure 9, different cross-sections of the same hailstone were selected to measure particle size distribution using CLSM and elemental composition via SEM-EDS, preserving the in-situ location relative to the embryo.

As seen in Figure 10-A,B, CLSM particle size distribution for V-7 reveals that particles range from 2 to 150 $\mu m$. In the V-7V cross-section (Figure 10-A), most particles have major axis lengths between 2 and 60 $\mu m$, whereas in the V-7H cross-section (Figure 10-B), most particles are relatively smaller, falling between 2 and 45 $\mu m$. Two additional hailstones of 8 $cm$ each (V-16 and V-17), collected from the same storm, were also analyzed and compared to V-7. The different axis ranges in this figure represent the varying number of particles analyzed and differences in size range from these differently sized hailstones, consequently, their analyzed cross-sections. Particles in both V-16 (Figure 10-C) and V-17 (Figure 110-D) tend to be larger compared to those in V-7. However, particles larger than 100 $\mu m$ were identified in all three hailstones collected from the same supercell.

CLSM also provides topographical and shape information about the analyzed particles, as shown in Figure 11, which is crucial for understanding ice nucleation processes in clouds; however, a detailed analysis of the topography and shape of the particles is reserved for a future publication focusing on applications of this method. Surface topography is an active site for ice nucleation, influencing nucleation modes and the energy barrier for ice formation. This aligns with findings from Holden et al. (2021), suggesting that surface topography plays a significant role in ice nucleation. Additionally, other laboratory studies (e.g., Gao et al., 2022) have demonstrated that particle shape, size, and coating can impact the ice nucleation ability of particles, including soot. Gao et al.'s paper shows that coatings or internal mixing have resulted in different ice nucleation abilities

compared to bare particles. Importantly, these studies emphasize that particle size is not the sole determinant, highlighting the critical role of surface topography and particle shape in influencing atmospheric ice nucleation.

To characterize the elemental composition of the non-soluble particles from the SEM-EDS analysis (e.g., Figure 7-B), a statistical-based clustering method was applied based on elemental weight percentages (e.g., Laskin et al., 2012). Prior to the analysis, the types of particles present in the hailstone were unknown, motivating the use of cluster analysis to identify the dominant element within each particle. Orange's 3 k-means clustering algorithm (Demšar et al., 2013). was used, providing silhouette scores of clustering results for various k values, where higher scores indicate better clustering. This approach enabled the identification of particle similarities based on statistical clustering. Subsequently, the predominant element within each category was determined based on the knowledge of each particle's cluster. This approach resulted in five unique categories: carbon-based (C-based), carbon-based with heavier metals (C-heavy), silicon-based (Si-based), silicon-based with heavier metals (Si-heavy), and chloride-based (Cl-based), as illustrated in Figure 11 for the three most common categories.

Particles included in the C-based group had a C abundance greater than 10% weight, with this abundance being higher than that of Cl and Si. Those categorized in the C-heavy group met the same criteria as the C-based group but also had an abundance greater than 1% weight of heavier metals such as Ti, Cr, Fe, Ni, Zn, Br, and Mo. Particles categorized in the Si-based group had a Si abundance greater than 10% weight, with this abundance being higher than that of C and Cl. The Si-heavy group met the same criteria but had an abundance greater than 1% weight of heavier metals such as Ti, Cr, Fe, Ni, Zn, Br, and Mo. Finally, particles with a Cl abundance greater than 10% weight, with this abundance being higher than that of C or Si, were categorized in the Cl-based group. Using this characterization framework, it was found that most particles in both cross-sections of V-7 were carbonaceous (C-based and C-heavy), followed by silicates (Si-based and Si-heavy) and salts (Cl-based), as shown in Figure 12.

A benefit of this method is the ability to isolate particles within the embryo compared to the outer layers. This allows the description of both the sizes and elemental compositions of particles that may have served in the nucleation process of the hailstone, similar to Michaud et al. (2014). Figure 13 isolates particles in the embryo regions of the V-7 cross-sections, as V-7V and V-7H cross through the embryo (see Figure 9 for reference). In the embryo region, salts were not identified in the initial analysis of V-7V (as seen in Figure 13-A). However, they were present in the additional horizontal cross-section (Figure 13-B; V-7H), along with heavier metals. Different particles were selected within the embryo sample to better elucidate the range of particle characteristics observed within this hailstone's embryo.

In summary, these findings highlight the robustness of our approach, revealing consistent overall messages while emphasizing the benefits of examining multiple cross-sections in a single stone. This method is particularly valuable for identifying a diverse range of elemental components. Within hailstones from the same storm, there appears to be a noticeable shift in particle size corresponding to the hailstone's increasing size, which is an interesting result to explore further. This unique method preserves particle size and elemental composition relative to the embryo, offering insights into the correlation between particle size, elemental distribution, and their origins in different cloud regions during hail formation. Such insights will contribute to a deeper understanding of nucleation properties and the source regions influencing hail formation and growth with the application of this technique to additional hailstones from various storm types and environments.

## 5 Limitations and Recommendations

This section discusses some limitations of the microscopy analysis technique, offering suggestions to mitigate these constraints and enhance the analysis by incorporating complementary techniques. A unique aspect of this work is the characterization of individual particles. While beneficial, it is also a time-consuming process. For instance, the analysis examined 176 particles using 9 hours of CLSM and 20 hours of SEM-EDS. There are pathways to consider when optimizing SEM-EDS analysis. Studies like those conducted by Lata et al. (2021) and Diep et al. (2022) have utilized a computer-controlled SEM-EDS (CCSEM) microscope for elemental chemical composition analyses, enabling the analysis of a larger number of particles while emphasizing interpretation. A computer-controlled "CC" SEM-EDS system incorporates additional software facilitating automated analysis, including automated sample loading, stage movement, focusing, imaging, and elemental analysis. CCSEM-EDS software can be programmed to autonomously analyze multiple particles consecutively without human intervention (Vander Wood, 1994). Utilizing CCSEM-EDS streamlines the analysis process and optimizes the funds allocated for this instrumentation.

Secondly, Section 3.3.2 noted the potential introduction of elements into the EDS spectral analysis from the substrate, in this case, glass. As previously mentioned, higher voltages provide a more accurate analysis of heavier elements but can introduce more contamination from the glass. The single-point measurement technique described in Section 3.3.2 provides a solution, albeit introducing more labor-intensive steps to the process. Two recommendations, therefore, emerge: 1) measuring particle-free areas of the glass using the EDS to obtain a control elemental spectrum from the glass or 2) decreasing the accelerating voltage to 8 $kV$ to reduce background interference and then rerunning the EDS analysis with a higher (e.g., 15 $kV$) voltage only on those particles showing the presence of heavier elements to increase accuracy in those composition measurements. Varying voltages in the EDS analysis in the context of this type of particle analysis are uncommon in the literature and may provide more confidence when characterizing non-soluble particles. Furthermore, to validate an individual particle's EDS spectrum result, spectrum mathematical operations were performed using the NIST DTSA-II program (Ritchie, 2010) to mitigate potential contamination originating from the glass substrate associated with a specific particle, as illustrated in Figure 14. The analysis revealed that it is possible to eliminate potential glass-related spectrum contaminants from the particle's spectral results (Figure 14-C) by subtracting the glass-related spectrum (Figure 14-B) from the particle-related spectrum (Figure 14-A). Spectral subtraction of particles from the underlying substrate enables more confident interpretations of element identity and abundance from non-soluble particles mounted on a glass substrate.

Due to the chemical composition of FORMVAR, the decision was made to quantify the contribution of C to the EDS spectral results when analyzing C-based particles using this technique. Potential C signals were compared in clear glass sections (i.e., not containing visible particles) with and without FORMVAR. Additionally, C signals associated with particles categorized as carbonaceous were examined to discern any significant differences in C presence. Results confirm minor contributions of C from FORMVAR, as illustrated in Figure 15. Due to its minimal thickness, the impact of the FORMVAR layer on the C signal is deemed negligible.

Further analysis using different microscopy techniques should be explored to more precisely categorize carbon-based particles into biological, organic, and inorganic categories. For example, Raman spectroscopy could aid the particle characterization

by complementing the SEM-EDS analysis, which does not provide enough information to discriminate different forms of Carbon in the sample. Raman Spectroscopy is a non-destructive chemical analysis technique that provides insights into a material's chemical structure, phase, polymorphy, crystallinity, and molecular interactions (Orlando et al., 2021). This technique yields unique chemical fingerprints, enabling swift identification and differentiation from other substances (Dahal, 2022). By combining Raman spectroscopy with complementary imaging methods, like SEM-EDS analysis used in this study, researchers could compare Raman distribution maps with topographical or morphological images, correlating Raman spectra with complementary SEM-EDS elemental information.

Additionally, scanning transmission X-ray microscopy (STXM) coupled with near-edge X-ray absorption fine structure (NEXAFS) spectroscopy can be explored to distinguish between organic and inorganic carbon (Lehmann et al., 2005; Moffet et al., 2011; Fallica et al., 2018; Lata et al., 2021). NEXAFS spectroscopy exploits the photoexcitation of electrons from a core level to unoccupied molecular orbitals to probe the specific chemical environment of a given element (Fallica et al., 2018). Mapping the distribution of organic carbon at the K-absorption edge of carbon could aid in further distinguishing between organic and inorganic carbon since STXM/NEXAFS applications have been used in atmospheric aerosol research (Moffet et al., 2011; Lata et al., 2021) and soil mapping research (Lehmann et al., 2005) to improve organic and inorganic carbon characterization.

## 6 Conclusions

Analyzing hailstone samples through the proposed methodology offers a promising avenue for gaining deeper insights into hailstones' composition, structure, and properties. This method has addressed and resolved several limitations observed in previous studies of collected hailstones by shifting the focus from melting to sublimating the hailstone, thus preserving the in situ particle distribution of all non-soluble particles within hailstones. This approach enables the identification of individual particles across each layer of the hailstone, including the embryo and air bubble rings. This approach, therefore, provides a precise representation of particle distribution and allows for the correlation of this data with the corresponding rings associated with specific growth zones within the cloud.

The integration of CLSM and SEM-EDS combines two powerful analytical techniques contributing to a comprehensive understanding of hailstones' microphysics and developmental processes. CLSM enables the characterization of non-soluble particles, revealing insights into particle size, surface topography, and early ice nucleation stages. SEM-EDS complements this by providing detailed elemental compositions, facilitating particle classification based on composition. Using a protective porous plastic coating preserves sample integrity, ensuring comprehensive analysis of particle spectra. These findings underscore the method's robustness in identifying diverse elements and revealing correlations between particle size, elemental distribution, and their origins in different cloud regions during hail formation. Applying these methods to hailstones from the same storm allowed us to isolate particle characteristics within the embryo, highlighting a noticeable shift in particle size with increasing hailstone size, providing opportunities for new insights into our understanding of nucleation properties and factors influencing hail growth.

While this proposed method offers significant advantages in providing more comprehensive data on the individual non-soluble particles trapped in a hailstone, it has some limitations. The optimization of human resources could be achieved by incorporating a CCSEM-EDS microscope, allowing for a more efficient allocation of time toward the interpretation of each individual particle's species. Additionally, while efforts have been made to minimize the impact of glass substrate background noise on EDS spectral analysis, caution should still be exercised in interpreting spectral results, especially for particles near the lower particle size limit.

The innovative approach outlined in this paper presents a valuable contribution to the field of hailstone analysis. By combining powerful and high resolution microscopy techniques and meticulous sample preparation, researchers can better uncover the intricate details of hailstone composition and properties, which can give insights into hail developmental processes. When applied to additional hailstones from different storm types and environments, this method will enhance our understanding of atmospheric particles' role in hailstone microphysics and contribute to advancements in weather forecasting, mitigation strategies, and climate change modeling.

## Appendix A: Cleaning protocol for quasi-clean sub-zero facilities

There are risks involved throughout the acquisition process and subsequent handling, leading to potential contamination and particle loss. Possible contamination sources during the processing of a hailstone may include the ultrapure water used, laboratory air, air from the dry container (silica gel dish), and the FORMVAR solution. The adapted sublimation particle separation procedure (described in Section 2.4) reduces these contamination sources compared to traditional methods that melt hailstones because the hailstone never transitions to the liquid phase; the diffusion coefficients of contaminants for the liquid phase (at $0°C, 4 \times 10^{-9} \ m^2/s$; Pruppacher, 1972) is at least one order of magnitude higher than in the solid phase (ice, $4 \times 10^{-15} \ m^2/s$; Petrenko and Whitworth, 1999). Nevertheless, a systematic approach is needed to identify and manage potential contamination sources, including the operator, the working environment, and the materials and tools employed in processing the sample until the particle deposit is completed and the sample is ready for further microscopy analyses.

The sub-zero facility used for the development of the microcopy method is composed of two rooms, a pre-chamber (entry chamber) and a working chamber; this approach is to minimize the introduction of possible contaminants when processing samples. Metal walls and floors were first cleaned with a detergent solution, rinsed with tap water, and disinfected with sodium hypochlorite (commonly used as a disinfectant and bleaching agent), using water with low-lint mops without typical cleaning chemicals, rinsed a second time with clean water, and later rinsed a last time with ultrapure water. The rooms do not have an air filter system. Due to the absence of a filtration system, given that the hailstone consists of ultra-pure water, it is particularly susceptible to absorbing airborne impurities, which could compromise sample integrity. However, due to the objectives of this method, this was not a concern at this time since this procedure is only meant to prevent contamination as much as possible without having to spend significant funds on chemical decontamination services, hence only achieving a quasi-clean room for hailstone preparation. All the equipment used to process thin sheets of ice was also washed with ultra-pure water. The glass slides were degreased with a non-ionic detergent and then rinsed multiple times with ultrapure water. The slides were then

dried in a clean, quasi-closed container at room temperature without a paper towel. Every operator used protective suits for the cold with an external waterproof fabric. The suits were washed using traditional detergent and rinsed multiple times using water alone. Each operator wore two gloves: the base was made of wool for protection from the cold, and a second latex glove was on top. Common plastic glasses for eye protection were used.

## Appendix B: Optimum thickness of the porous plastic coating layer

Beyond the risk of introducing contaminants into the substrate, certain particles might move from their original location during handling or in coating procedures affected by vacuum pressures needed for SEM-EDS analyses. As previously mentioned, a way to address this limitation is to apply a FORMVAR solution film over the sample. Determining the ideal plastic coating layer thickness is essential to using the adapted sublimation method effectively described in Section 2.2. Achieving the right mixture of FORMVAR and ethylene dichloride is essential; the layer must be sufficiently thick to trap particles in the hailstone while remaining thin enough to reduce the sublimation duration (24 to 48 hours and not weeks). Additionally, it must be thin enough to allow the SEM electron beam to penetrate the plastic coating to analyze particles, as seen in Figure 1.

Because the FORMVAR is transparent, the plastic layer thickness presents no major limitations in analyzing particles in a sublimated hailstone sample with the CLSM. However, the thickness limitation is of greater concern when preparing the sample for SEM-EDS analysis to determine the elemental chemical composition of particles. For samples to be analyzed using the SEM, they must be coated with a layer of electrically conductive material, typically with a layer of a metal coating, with a thickness of an order of 25 $nm$ in a vacuum environment of $5 \times 10^{-2}$ $mBar$. Additionally, after coating, the sample is inserted into a chamber in the SEM instrumentation that undergoes a vacuum of $5 \times 10^{-5}$ $mBar$ or greater. If no plastic coating was applied to the sublimated hailstone sample, the vacuum process in both previously described scenarios could remove particles, which makes it essential that the sample is covered with a protective film that keeps the particles trapped.

Multiple lab-grown ice samples were prepared and coated with one to three layers of FORMVAR solution at 1 and 3 % concentrations to address the thickness considerations and minimize particle displacement. This procedure consists of sweeping the polished surface of the ice sample with a rod dipped in FORMVAR solution. The sample is then analyzed with CLSM, where an area exhibiting a distinct FORMVAR relief pattern is identified within the sample, as seen in Figure B-1 (top). Then, a transect across this pattern is used to measure the height of the relief (Figure B-1, bottom) and thus determine the thickness of the FORMVAR layer. Thicknesses measured for multiple FORMVAR coatings are shown in Table B1. This approach determined that one layer of coating at 1 % FORMVAR thickness optimally addressed the aforementioned tradeoffs of the plastic coating, i.e., preserving particles while being able to see particles with the SEM (e.g., Figure 1).

| FORMVAR Concentration | Number of **FORMVAR** layers | | |
|---|---|---|---|
| | 1 | 2 | 3 |
| | Layer thickness in $[\mu m]$ | | |
| 1% | 0.007 | 0.031 | 0.100 |
| 3% | 0.7 | 1.7 | 30.0 |

**Table B1.** FORMVAR coating thicknesses measured with CLSM

*Author contributions.* Conceptualization, A.C.B.A. and L.E.A.; methodology, A.C.B.A., A.K.R. and L.E.A., M.L.A.; validation, A.K.R. and L.E.A., W.O.N.; formal analysis, A.C.B.A., and L.E.A.; investigation, A.C.B.A., A.K.R. and L.E.A.; resources, A.K.R. and L.E.A.; data curation, L.E.A.; writing—original draft preparation, A.C.B.A.; writing—review and editing, A.C.B.A., A.K.R., L.E.A. and W.O.N., M.L.A.; visualization, A.C.B.A.; supervision, A.K.R., L.E.A. and W.O.N.; funding acquisition, A.K.R.. All authors have read and agreed to the published version of the manuscript.

*Competing interests.* The authors declare no competing interests.

*Acknowledgements.* This research was funded by the National Science Foundation (grant no. AGS-1640452, AGS-1661768). We thank Sebastian García at the Laboratorio de Análisis de Materiales por Espectrometría de Rayos X" facilities at the Universidad Nacional de Cordoba for his technical assistance in the microscopy analysis phase. We also extend our gratitude to the reviewers, whose insightful comments have significantly improved the quality of this paper.

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

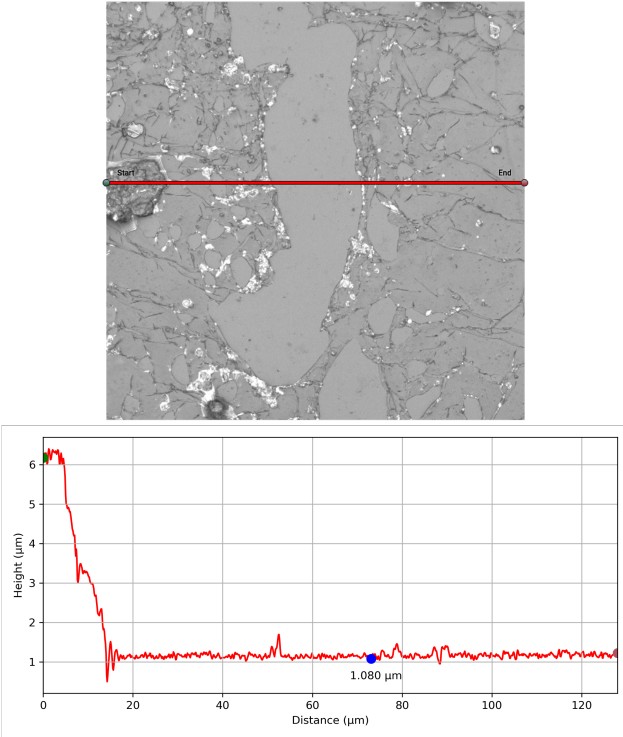

**Figure B1.** A snapshot from the Olympus LEXT Analysis software of a sublimated ice sample with FORMVAR coating. The top panel shows the 2-D CLSM laser imagery with the red line indicating the transect through an area with and without the coating. The bottom panel shows the height (y-axis in $\mu m$) across that transect (x-axis in $\mu m$) for calculating FORMVAR thickness (as seen along the blue segment), plotting the height used for the FORMVAR's thickness calculation shown in the blue dot.

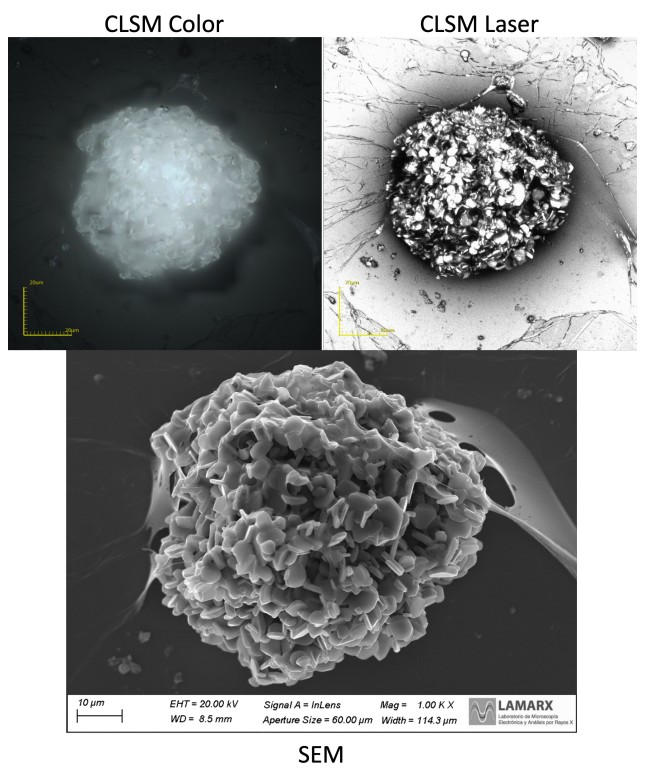

**Figure 1.** A particle trapped in a hailstone as observed by CLSM and SEM for 1% FORMVAR coating.

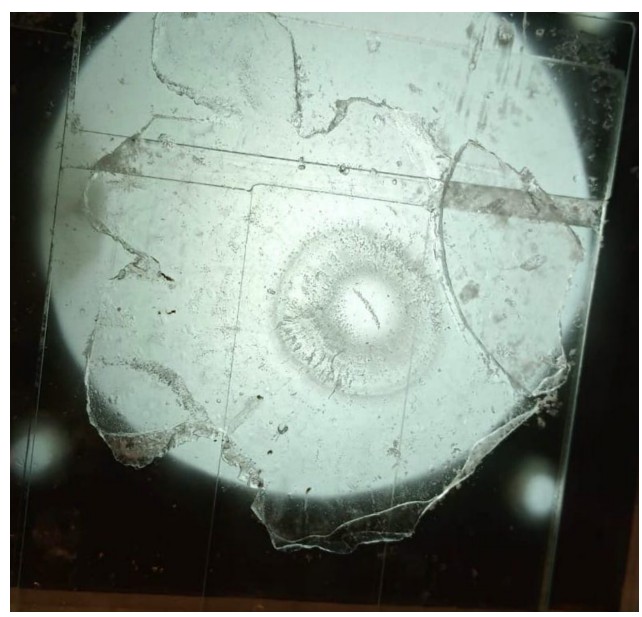

**Figure 2.** An example of a thin cross-section of a hailstone sectioned through its equatorial plane. It is affixed to several thin glass sections arranged in a mosaic pattern on a thin glass slide with dimensions of 25 x 75 $mm$.

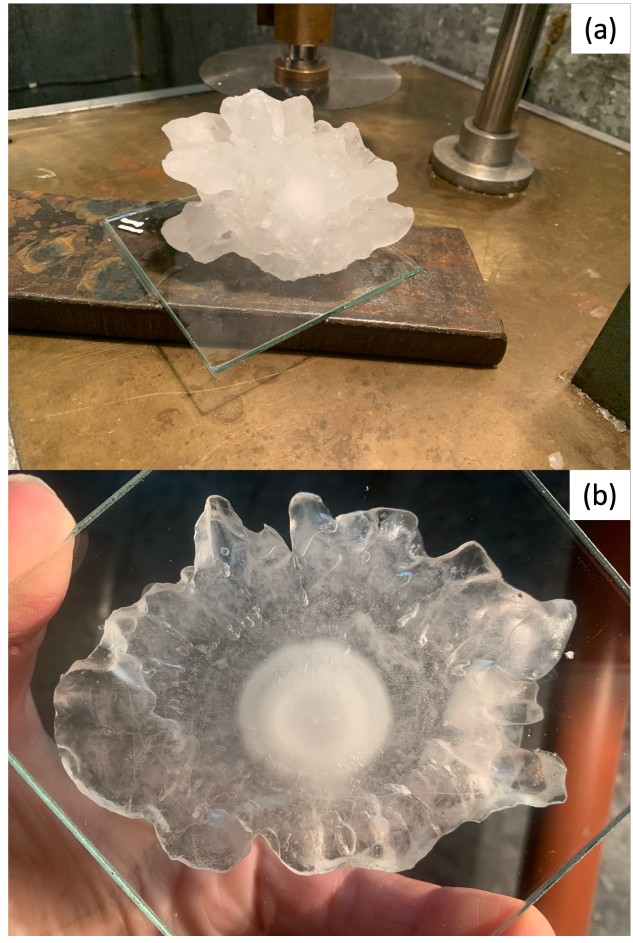

**Figure 3.** The figure shows the preparation stages of a hailstone. (a) The hailstone is attached to a glass section and then to a metal section plate with ultra-pure liquid water. (b) The hailstone is cut in the equatorial symmetric plane to provide a view of the hailstone's embryo.

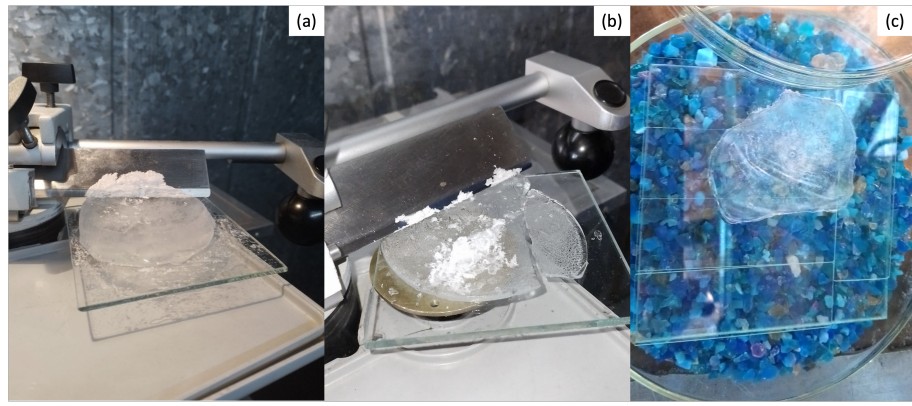

**Figure 4.** Figure showing before (a) and after (b) a hailstone is evenly polished to the desired thickness with a microtome to obtain a top view of the embryo, and (c) is left to sublimate in a humidity-controlled container with silica gel.

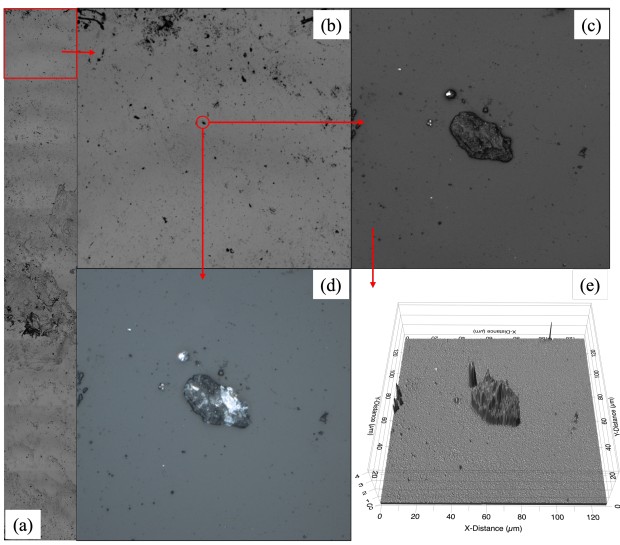

**Figure 5.** Example CLSM imagery from the coated, sublimated hailstone shows (a) a 2-D cross-section along an axis in the equatorial plane at 108x magnification, (b) a subsector scanned at higher magnification for creating (c) visible and (d) laser images for individual particles. (e) 3D particle surface topography from the visible imagery is analyzed with ProfilmOnline software.

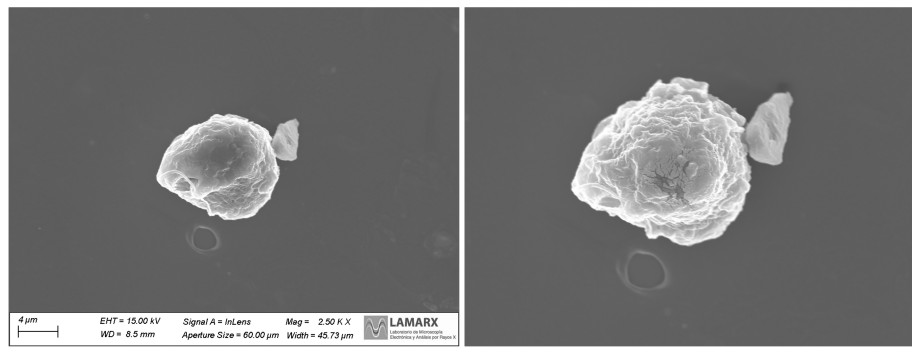

**Figure 6.** The left panel displays the particle prior to undergoing EDS analysis, while the right panel illustrates the particle post-EDS analysis, revealing subtle breakage and disintegration concentrated at the center of the particle where the SEM-laser was centered.

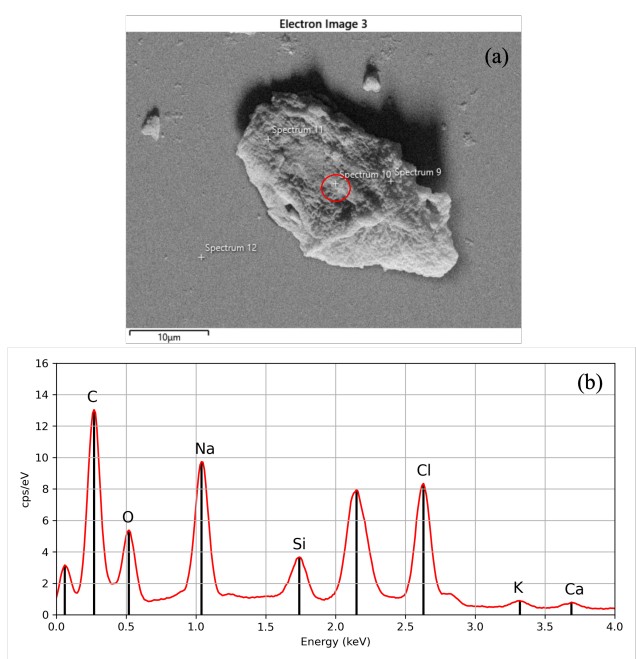

**Figure 7.** (a) A SEM backscattered electron image for an individual particle categorized as carbon-based with (b) the corresponding EDS elemental composition for that particle. The first peak, closest to 0 keV, is caused by noise from the EDS detector and can be ignored when analyzing the elemental composition of a particle, as it does not represent any actual element. The second spectral peak corresponds to gold (2.12 keV) and was excluded from the analysis since it was used for coating the sample.

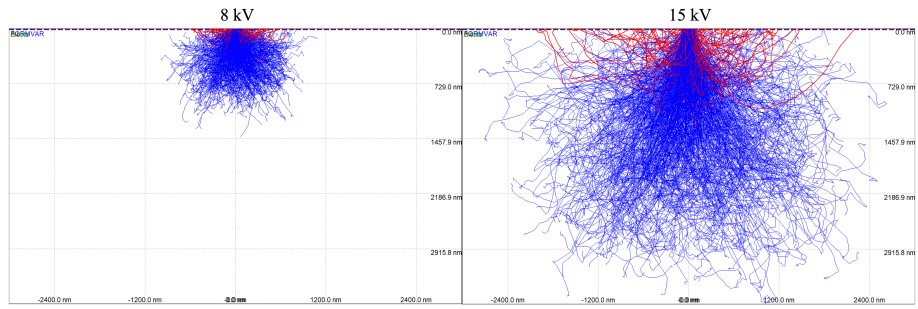

**Figure 8.** Activation volume producing Fe X-rays within the soda-lime glass for 8 (left) and 15 $kV$ (right) accelerating voltage from the Monte-Carlo simulations. Lines indicate the pathways of individual electrons backscattered from within the sample (blue) and from the sample surface (red) as a function of particle size (in $nm$).

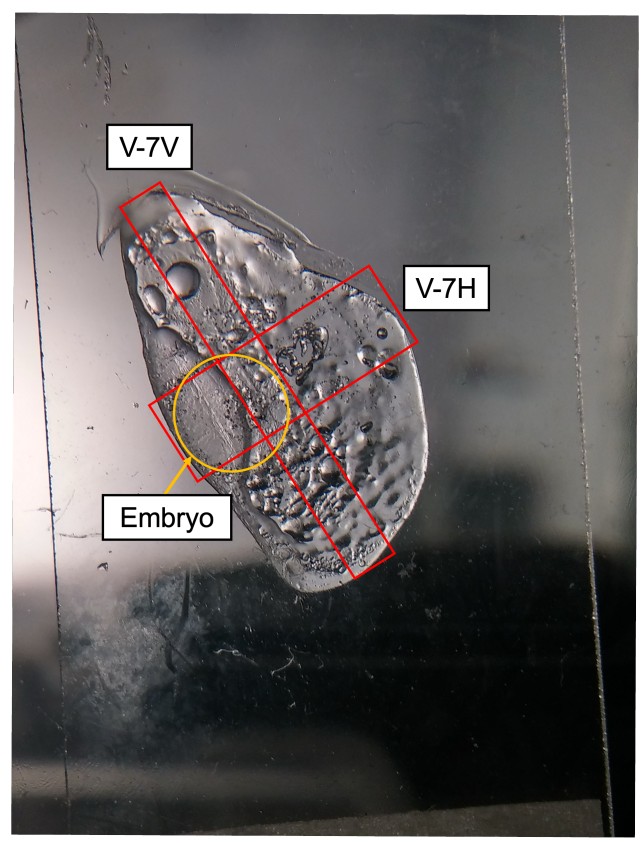

**Figure 9.** An example analyzed hailstone (V-7) where the areas highlighted by red rectangles indicate where particles were randomly selected to measure particle size distribution using CLSM and elemental composition via SEM-EDS. The orange circle marks the location of the embryo.

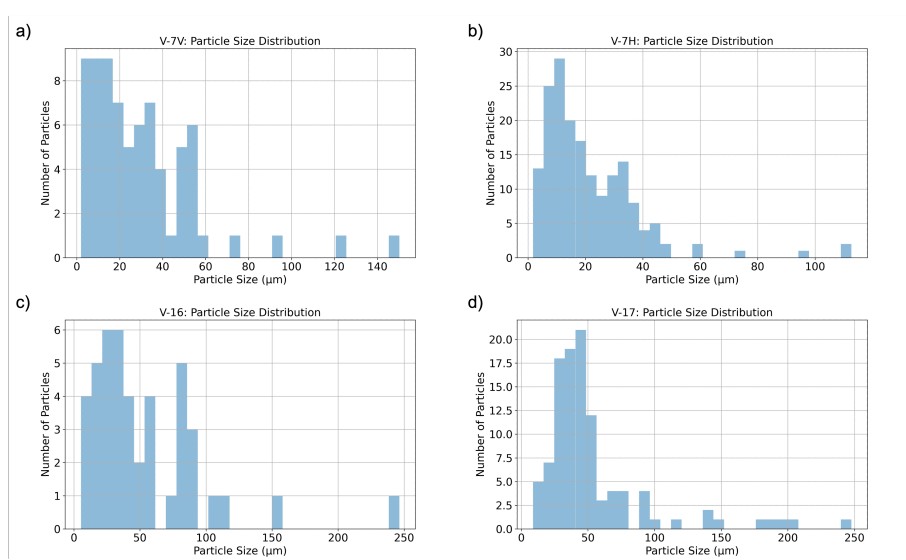

**Figure 10.** Particle size distributions for hailstone samples collected during the event on 8 February 2018. Panels a) and b) display particle sizes for two different cross-sections from sample V-7, while panels c) and d) show particle sizes for samples V-16 and V-17, respectively.

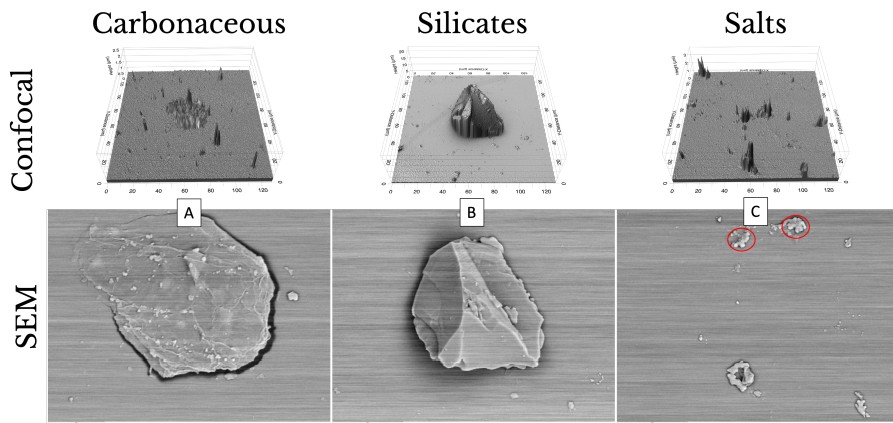

**Figure 11.** Examples from each of the primary particle categories are presented, along with their ProFilmOnline topographical output obtained through a Confocal Laser Scanning Microscope (CLSM, top), accompanied by their corresponding Scanning Electron Microscopy (SEM, bottom) image.

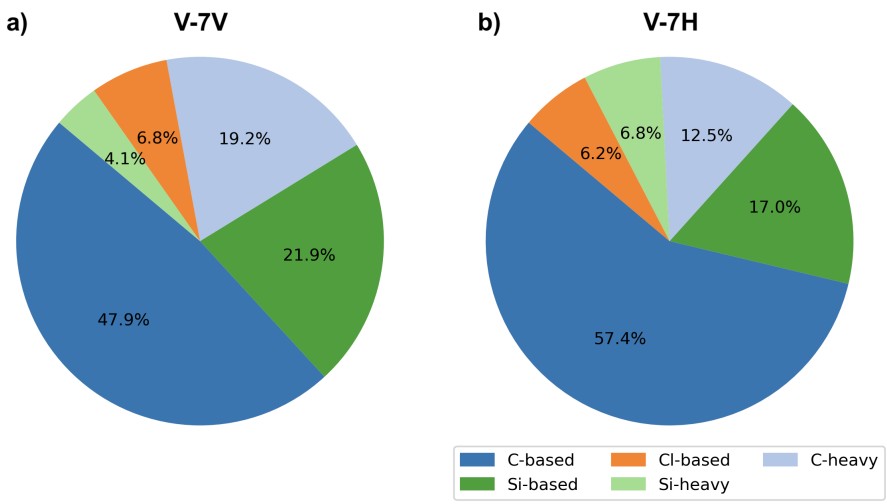

**Figure 12.** EDS-based elemental composition distribution of particles for a) V-7V and b) V-7H cross-sections, as seen in Figure 9.

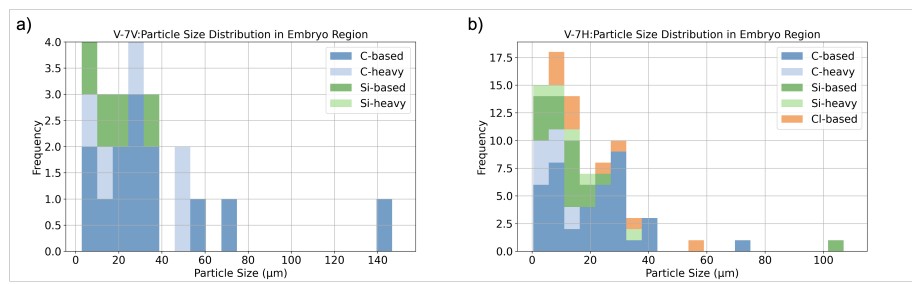

**Figure 13.** EDS-based elemental composition distribution of particles found in the embryo for sample V-7 for a) V-7V and b) V-7H.

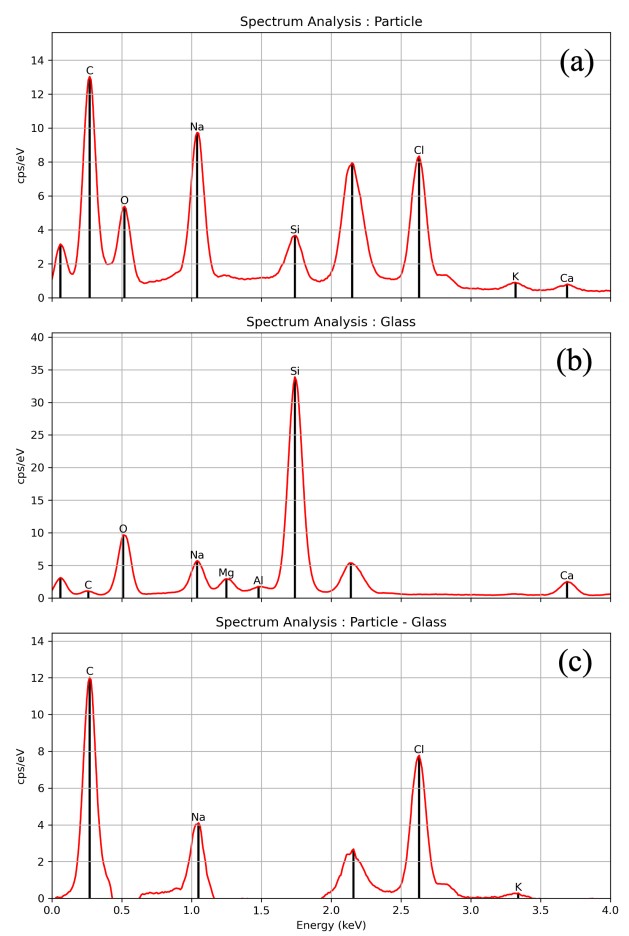

**Figure 14.** Mathematical operations on the spectrum using DTSA-II were done to analyze the EDS-spectrum data and differentiate a particle's spectrum from the glass substrate. a) Depicts the spectrum of the particle presented in Figure 7-B, b) illustrates the spectrum of the glass, and c) demonstrates the outcome of the subtraction, revealing the presence of elements C, Na, Cl, and K in the particle. The first peak, closest to 0 keV, is caused by noise from the EDS detector and can be ignored when analyzing the elemental composition of a particle, as it does not represent any actual element. The second spectral peak corresponds to gold (2.12 keV) and was excluded from the analysis since it was used for coating the sample.

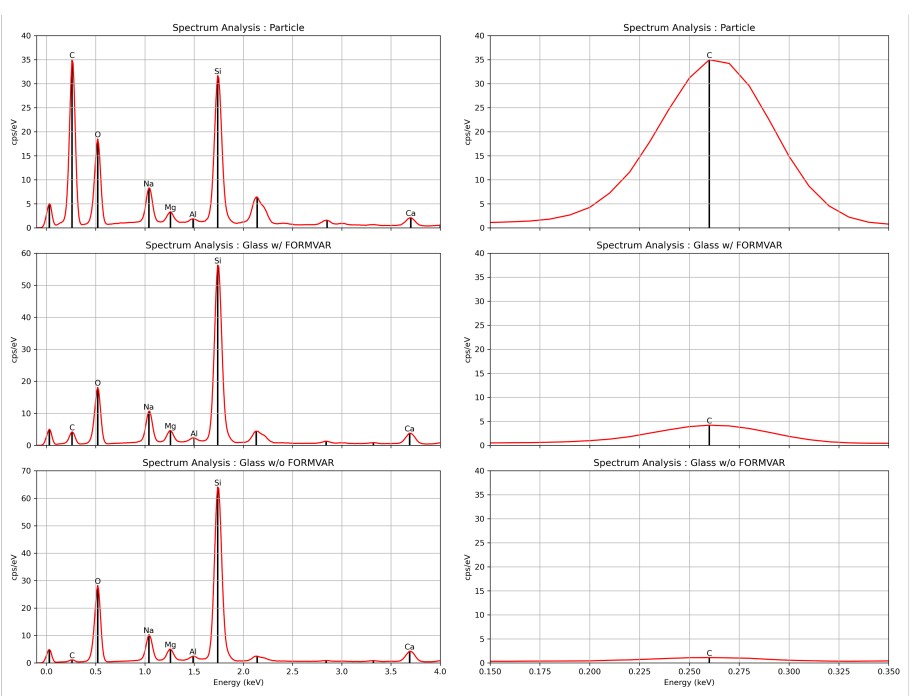

**Figure 15.** The left figure displays a single-point spectral analysis of a C-based particle with FORMVAR (top panel), a clear area with FORMVAR (middle panel), and a clear area without FORMVAR (bottom panel). Each panel in the figure to the right is the corresponding panel providing a closer view of the C peak. Due to the FORMVAR's thickness and the results in this figure, the FORMVAR layer's impact on the C peak is considered negligible. The first peak, closest to 0 keV, is caused by noise from the EDS detector and can be ignored when analyzing the elemental composition of a particle, as it does not represent any actual element. The second spectral peak corresponds to gold (2.12 keV) and was excluded from the analysis since it was used for coating the sample.