# Peer review of "Exploring Non-Soluble Particles through Innovative Confocal Laser and Scanning Electron Microscopy Techniques"

_EGUsphere, 2023_

## Author Comment (AC1)

**1.    Although they focus on insoluble particles, as indicated in the title, I believe the authors measured sea salt, soluble particles. The composition shown in Figure 7 shows a strong presence of Na and Cl. The result may indicate that they measured crystallized particles that were once soluble in the hailstones or their precursors. I believe that the result is still good, but some discussion needs to be clarified including the title.**

Thank you for your insightful observation. This paper briefly mentioned that during sublimation, dissolved components may undergo precipitation. For instance, in cases where the original hailstone possessed a brine-like composition, the sublimation process could result in the precipitation of once-soluble salt particles.

However, in convective systems, sodium chloride and other crystalline inorganic salts injected into an updraft have the potential to remain undissolved in the surrounding water, acting as ice-nucleating particles under certain conditions. Research has highlighted the ability of deliquescent salts to serve as INPs in the atmosphere (1,2). Additionally, soluble salt particles like NaCl, NaI, KI, and KCl have been found to induce contact freezing at specific temperatures, further supporting their role as INPs (3).

The behavior of salt crystals in a convective system is influenced by multiple factors, such as solution properties and convective transport processes (e.g., updraft speed). There's a possibility that salt crystals may act as INPs without dissolving in the surrounding water. However, the unique contribution of this paper lies in its ability to analyze insoluble particles, acknowledging that soluble particles are not explicitly ruled out. Nonetheless, the focus remains on insoluble particles, and this distinction will be discussed further in the text without the need to change the title.

(1)      https://acp.copernicus.org/articles/21/13903/2021/
(2)      https://agupubs.onlinelibrary.wiley.com/doi/full/10.1002/2017JD027864
(3)      https://journals.ametsoc.org/view/journals/amsm/58/1/amsmonographs-d-16-0006.1.xml

**2.    I think Figure 9 is the most important result in this study, but they simply classify the particles into three categories: C-, Cl-, and Si-based.  I suggest they discuss their particle categories more. I assume they are carbonaceous, sea salt, and dust particles. With particle shape analysis and more detailed compositional analyses, it would be possible to identify them. At least for simple classifications such as those in Fig. 9, there is no need to discuss SEM-EDS conditions, but a rough qualitative measurement can classify them.**

We acknowledge the importance of Figure 9 in our study and appreciate the suggestion to provide more detailed discussions on particle categories. In categorizing unknown particles within hailstones, we employed k-means clustering and silhouette scores for statistical clustering to reveal particle similarities (Demšar et al., 2013) (1). Subsequently, we identified three main clustered groups based on the predominant elements C, Si, and Cl. These elements were

determined by analyzing each particle cluster and determining the dominant element in each cluster.

The reviewer's assumption aligns with our findings, as the predominant elements generally correspond to carbonaceous, dust, and salt particles. A similar correlation was previously observed by Lata et al. in 2021 (2). Their paper inspired our approach to create a decision flow chart to set elemental percentage thresholds to separate the particles into the categories found through the k-means clustering approach. However, it's important to note that carbonaceous particles may also originate from sources like soil dust, referred to as soil organic carbons, which are particularly common in arid and semi-arid areas (3). Furthermore, our analysis found instances where both silicates and carbonaceous particles exhibited the presence of heavier metals.

We've included an illustrative example in the figure below to show a representative particle from each identified category, allowing for more precise differentiation under CLSM and SEM. Hu et al. (2022) (4) is an example of a study that used the SEM to examine dust particles. By comparing the morphology of the dust particles in their study, for example, with images obtained through our study's method, we can draw comparisons with other particles previously studied using SEM. To emphasize the analytical capabilities of the methods employed in our study, including studying dust morphology, we will include the figure below in the revised text showing the confocal topography and SEM images for our different particle types.

[Figure]

(1) https://orangedatamining.com/
(2) https://pubs.acs.org/doi/10.1021/acsearthspacechem.1c00315
(3) https://www.mdpi.com/2073-445X/11/2/176
(4) https://www.sciencedirect.com/science/article/pii/S0048969722024081

**3. In the later discussion, there is very limited discussion of the results from the CLSM. I suggest more discussion using the results.**

We acknowledge the need for a more comprehensive discussion regarding the results obtained from the CLSM. We emphasized how we used it to locate the particle and ensure it was the same particle being analyzed in the SEM. However, CLSM also provides topographical and shape information about the analyzed particles, as seen in the figure in our previous response (which will be included in the updated version of the manuscript). While a detailed analysis of the topography and shape of the particles is for a future publication focusing on applications of this method, this information does have implications for understanding ice nucleation processes in the cloud.

Surface topography is an active site for ice nucleation, influencing nucleation modes and the energy barrier for ice formation. This aligns with findings from Holden et al. in 2021 (1), suggesting that surface topography plays a significant role in nucleation. Additionally, other laboratory studies (2) have demonstrated that particle shape, size, and coating can impact the ice nucleation ability of particles, including soot. This paper shows that coatings or internal mixing have resulted in different ice nucleation abilities compared to bare particles. Importantly, these studies emphasize that particle size is not the sole determinant, highlighting the critical role of surface topography and particle shape in influencing atmospheric ice nucleation.

In our upcoming revision, we will incorporate a more thorough discussion of the CLSM results within the context of the figure discussed in the previous answer.

(1) https://www.pnas.org/doi/full/10.1073/pnas.2022859118
(2) https://acp.copernicus.org/articles/22/5331/2022/

**4. Although the authors have many discussions about the influence of glass plates, I suggest using a metal plate or a substrate consisting of single elements (e.g., Cu, Al, and C). A glass plate contains so many elements that it interferes with the particle composition. By using other substrates, you will reduce such interferences.**

We appreciate your comment. Initially, we considered using a nickel base, as we anticipated finding elements in the hailstone, such as Al and C, based on previous studies (e.g., Michaud et al. 2014). However, using a glass base, we can examine the sample using transmitted light microscopes (i.e., CLSM), which would not be possible with a metal substrate, as glass is the only substrate transparent to visible light. Additionally, we encountered challenges with adhering hailstones to a metal surface, as we have yet to experiment to assess the adherence of ice to the nickel or other metal bases and the potential reactions of formvar during the sublimation process.

Given our need for using the CLSM, plus our exploration of various methods to prevent contamination, including methods to remove/subtract such contamination from spectral results,

we have found that the current approach using glass is the most suitable for extracting the information we seek from the hailstones.

**5.    Did formvar coatings also contribute to the C signal? If so, C-based particles may need to be reconsidered.**

Good question. We compared potential C signals in clear glass sections (i.e., not containing visible particles) with and without formvar. Additionally, we examined C signals associated with a particle categorized as a carbonaceous to discern any significant differences in C presence. Our findings confirm minor C contributions from the formvar, as illustrated in the figure below. Due to its minimal thickness, the formvar layer's impact on the C signal is deemed negligible. We value this observation and will incorporate this discussion into the appropriate section of the revised text.

[Figure]

The left figure displays a single-point spectral analysis of a C-based particle with formvar (top panel), a clear area with formvar (middle panel), and a clear area without formvar (bottom panel). Each panel in the figure to the right is the corresponding panel providing a closer view of the C peak. Due to the formvar's thickness and the results in this figure, the formvar layer's impact on the C peak is considered negligible.

**6.   In the conclusion section, I do not think it is good to introduce other techniques such as Raman spectroscopy and STXM. They could be placed in the recommendation or other sections.**

Thank you for your recommendation. I agree with the suggestion, which will be addressed in the subsequent version of this paper.

**7.   Could biological particles be identified by SEM using their specific morphology and composition, such as a tracer of P, S, and Cl?**

The SEM is a valuable tool for identifying biological particles based on their unique morphology and elemental composition. This imaging technique provides high-resolution, three-dimensional images of specimen surfaces, enabling the visual characterization of known biological structures such as cells, bacteria, and viruses. This can be coupled with EDS to analyze elemental tracers like phosphorus, sulfur, and chlorine.

However, a known biological particle must be used as a control for accurate comparisons with potential traces found in unknown particles within a hailstone. Our hailstone analysis method was initially developed using a hailstone sample that exhibited relatively weak traces of phosphorus and sulfur, rendering these elements unsuitable as references for distinguishing biological particles.

EDS, in general, cannot distinguish organic from inorganic carbon; this method encourages the exploration of supplementary methods (e.g., Kirchstetter et al., 2004 (1), Moffet et al., 2011 (2), Orlando et al., 2021 (3)) to enhance the overall understanding of biological particles, especially in scenarios involving subtle traces of specific elements.

(1) https://agupubs.onlinelibrary.wiley.com/doi/full/10.1029/2004jd004999
(2) https://digital.library.unt.edu/ark:/67531/metadc831481/
(3) https://doi.org/10.3390/chemosensors9090262

**1.   Line 80: "Energy dispersive spectroscopy" should read "Energy dispersive X-ray spectroscopy".**

Thank you for this comment. This will be addressed in the revised manuscript.

**2.   Line 83: SEM works by scanning a focused beam of electrons.**

I propose the following correction, which will change the text from "SEM works by focusing a beam of electrons onto the sample, which causes the emission of secondary electrons and backscattered electrons." to "SEM works by scanning a focused beam of electrons onto a sample, inducing the emission of secondary electrons and backscattered electrons."

**3.    Line 108: I think a sublimation point depends on both humidity and temperature, whereas a melting point depends only on temperature.**

Thank you for your comment. Both sublimation and the melting point of water depend on temperature and partial vapor pressure (which is proportional to humidity), as shown in the following figure. For this reason, we consider that no change is needed to the text.

[Figure]

**4.    Line 170: I was not sure if sublimation in dry air can occur with silica gel. Although it can be determined by a detailed calculation, I think silica gel may not achieve the low humidity needed for sublimation. I am not sure about the current conditions, but it is better to check.**

We argue that the silica gel can achieve the low humidities required for sublimation because the hailstone sample from Central Argentina used in this case successfully underwent sublimation within a 48-hour timeframe. We are unsure what "current conditions" means and are open to discussing this point further.

**5.    Line 197-200: Although a high vacuum SEM has a better SEM image than a low vacuum SEM, a low vacuum SEM has sufficient EDS capability for the purpose used in this study.**

Thank you for highlighting this aspect. Not all SEM models incorporate low vacuum environment capability, and during the development of our method, the SEM at our disposal lacked this feature.

Many SEMs with this capability to analyze samples with low vacuum conditions introduce a known gas, like nitrogen, or allow some oxygen from the environment to enter the chamber. However, this will raise potential concerns. One concern is contamination from the surrounding air. Without a clean oxygen source entering the chamber or a filter for purification (because SEMs do not have built-in air filtration systems), contaminants from the surrounding atmosphere could compromise the chamber. The hailstone consists of ultra-pure water, so it is susceptible to absorbing impurities in the air, which could impact the sample's integrity.

Another consideration under low vacuum conditions is atmospheric skirting, wherein a gaseous environment modifies the primary electron beam profile. The electron beam is typically divided into two fractions: an un-scattered beam with the original distribution profile and diameter and a scattered beam forming a "beam skirting" around it (1,2,3). This beam alteration occurs before reaching the particle surface, impacting the resolution of high-resolution imagery and spectral analysis through the EDS.

Therefore, due to potential issues related to contamination, lack of control over the beam's trajectory, and the resulting impact on resolution and spectral analysis, we respectfully disagree with the notion that low vacuum conditions would suffice for our method. We appreciate this observation; this discussion will be included in the following version.

(1) https://www.sciencedirect.com/science/article/pii/S0065253908609026
(2) https://onlinelibrary.wiley.com/doi/abs/10.1002/sca.4950230505
(3) https://onlinelibrary.wiley.com/doi/abs/10.1002/sca.4950220304

**6.  Line 204: Is "sigma" OK?**

Thank you for this observation; it will be updated in the following version to "Sigma."

**7.  Line 208-209: A high voltage does not always improve the spatial resolution of SEM images due to its expansion in the materials. Please check again.**

Thank you for your comment. While the statement in question is supported by literature (Goldstein et al., 2017) (1), I agree with the reviewer that there are instances where excessively high voltages may lead to the expansion or penetration of electrons into the materials, potentially affecting the resolution negatively. I proposed to rewrite the sentence in the following way:

"The accelerating voltage of the primary beam determines the wavelength of electrons, and higher voltages are generally advised for enhanced spatial resolution in electron imaging (Goldstein et al., 2017). However, it's important to consider that this principle may not universally apply to all materials. In some cases, excessively high voltages could result in electron expansion or penetration into the materials, potentially diminishing resolution. Therefore, it is essential to determine the optimal voltage, potentially opting for a lower one, when analyzing specific samples to ensure optimal imaging resolution."

(1) https://link.springer.com/book/10.1007/978-1-4939-6676-9

**8.   Line 214: I believe that a working distance does not affect the beam diameter. Please check it.**

The relationship between working distance and beam diameter in SEM systems is well-established in existing literature (1,2). The working distance, representing the distance between the final lens of the SEM column and the specimen, influences the beam diameter, which is the diameter of the electron beam at the specimen. The electron optics within the SEM column governs this relationship.

As detailed in literature reference (1), the increase in working distance results in a proportional increase in the beam diameter. The geometric spreading of the electron beam over a greater distance from the final lens to the specimen contributes to this phenomenon.

Moreover, studies on the design and fabrication parameters for optical systems, including SEM (2), emphasize the interdependence of working distance and beam diameter. Specifically, variations in the working distance directly impact the beam diameter, with an increase in working distance corresponding to a larger beam diameter. This reinforces our statement in the paper, aligning with established literature on SEM systems.

(1)     https://www.sciencedirect.com/topics/engineering/lateral-resolution
(2)     https://www.mdpi.com/1424-8220/18/12/4150

**9.   Line 225: A 15 kV can measure higher than Fe. Please check it.**

The sentence reads: "The choice of 15 kV ensures that heavier elements are included in the EDS analysis by exciting the K lines of elements up to Fe." However, the intention was not to imply that we can measure up to Fe at a 15 kV accelerating voltage; instead, the emphasis is on the capability to measure heavier elements such as Fe at this voltage. It will be addressed in the revised version.

**10.   Line 237: There is no Figure 7D.**

Thank you for pointing out this discrepancy. In the next version of this paper, the reference will be corrected to "7-B," aligning with the intended figure for citation in this text.

**11.   Line 239: I agree. Please see my major comment**

Thank you for your feedback. We acknowledge your agreement and have considered your major comment.

**12.   Line 258-260: If the particles have been classified according to these criteria, there is no need to use the cluster analysis (line 253-257).**

Thank you for the opportunity to clarify our classification approach. Going into the study, we did not know what types of particles we would find in our hailstone, so we started with a cluster analysis to see which elemental clustering was dominant in our sample. Utilizing k-means and silhouette scores analysis allowed us to identify particle similarities based on statistical clustering. Subsequently, with the knowledge of each particle's cluster, we determined the predominant element within each category (i.e., carbonaceous, silicates, etc.). Applying thresholds derived from our data (i.e., C > 10% meant a "dominant" element), we categorized each particle within the defined groups: Carbonaceous (C greater than 10% and greater than Si and Cl), Silicates (Si greater than 10% and greater than C and Cl) and Salts (Cl greater than 10% greater than C and S).

Determining how the data should be categorized without this initial clustering would be challenging. Therefore, cluster analysis was necessary to ensure an appropriate classification of particles. We will emphasize this point in the revised manuscript.

**13.   Line 275: Is it true that CCSEM can load samples automatically?**

Based on the information provided by SEM manufacturers, computer-controlled scanning electron microscopes offer automated sample-loading capabilities (1,2,3,4,5). For instance, the ZEISS EVO scanning electron microscope incorporates Automated Intelligent Imaging, a feature that enhances sample throughput. Additionally, it provides tools for relocating regions of interest and ensuring the integrity of collected data, thus facilitating automated and efficient sample handling. Similarly, the JEOL FE-SEM has an AI system called NeoEngine, which tracks electron beam trajectories, streamlining operations with minimal user intervention. We will add a reference to the revised paper to describe this capability.

(1) https://www.zeiss.com/microscopy/en/products/sem-fib-sem/sem/evo.html
(2) https://www.azom.com/article.aspx?ArticleID=19595
(3)https://www.thermofisher.com/us/en/home/electron-microscopy/products/desktop-scanning-electron-microscopes.html
(4)https://www.semtechsolutions.com/blog/better-performance-from-scanning-electron-microscope-use/
(5) https://www.hitachi-hightech.com/global/en/sinews/technical_explanation/130301/

**14.   Line 276-277. I do not believe that CCSEM can measure thousands of particle compositions in less than an hour. EDS needs an acquisition time of at least several seconds.**

After reviewing your comment and cross-referencing it with my notes, I agree with your observation. More precisely, when we focused on analyzing individual particles, as described in

this methods paper, we assigned a 2-minute acquisition time. Considering this time constraint, our analysis permitted the theoretical examination of approximately 30 single-point analyses per hour. These analyses could be for a single particle or encompass multiple spots within a single particle. To accommodate your observation, I propose that the sentence be rewritten: "A CCSEM-EDS software can be programmed to autonomously analyze multiple particles consecutively, without requiring human intervention (Vander Wood, 1994)."

**15.   Line 280: There is no section 2.4.2.**

Thank you for identifying this inconsistency; this will be updated in the next version to "section 3.3.2."

**16.   Line 285-288. I do not think changing the acceleration voltage is effective. First, when using low voltage first, you will not see heavy elements. Second, it is very time-consuming, as suggested in line 278.**

Adjusting the voltage, although time-consuming, becomes especially relevant when analyzing particles that are 1 micron or smaller in size. This is because the SEM has a higher resolution compared to the CLSM. This higher resolution in the SEM gives us a distinct advantage, allowing for a more detailed examination of smaller particles. This prevents any elemental contribution from the base due to a large activation volume, as illustrated in Figure 8 and discussed in depth surrounding this figure.

**17.   Figure 7. There is a missing peak identification around 2.1 keV. Why is this?**

The spectral peak corresponds to gold, as it was used in coating the sample and is consequently excluded from the analysis. I will incorporate this clarification into the figure caption.

**18.   Figure 9. I cannot see the right images. Are they from Figure 6?**

Thank you for identifying this inconsistency; this will be updated in the next version to "... Figure 5."

---

## Author Comment (AC3)

**1.   Although they focus on insoluble particles, as indicated in the title, I believe the authors measured sea salt, soluble particles. The composition shown in Figure 7 shows a strong presence of Na and Cl. The result may indicate that they measured crystallized particles that were once soluble in the hailstones or their precursors. I believe that the result is still good, but some discussion needs to be clarified including the title.**

Thank you for your insightful observation. This paper briefly mentioned that during sublimation, dissolved components may undergo precipitation. For instance, in cases where the original hailstone possessed a brine-like composition, the sublimation process could result in the precipitation of once-soluble salt particles.

However, in convective systems, sodium chloride and other crystalline inorganic salts injected into an updraft have the potential to remain undissolved in the surrounding water, acting as ice-nucleating particles under certain conditions. Research has highlighted the ability of deliquescent salts to serve as INPs in the atmosphere (1,2). Additionally, soluble salt particles like NaCl, NaI, KI, and KCl have been found to induce contact freezing at specific temperatures, further supporting their role as INPs (3).

The behavior of salt crystals in a convective system is influenced by multiple factors, such as solution properties and convective transport processes (e.g., updraft speed). There's a possibility that salt crystals may act as INPs without dissolving in the surrounding water. However, the unique contribution of this paper lies in its ability to analyze insoluble particles, acknowledging that soluble particles are not explicitly ruled out. Nonetheless, the focus remains on insoluble particles, and this distinction will be discussed further in the text without the need to change the title.

(1)      https://acp.copernicus.org/articles/21/13903/2021/
(2)      https://agupubs.onlinelibrary.wiley.com/doi/full/10.1002/2017JD027864
(3)      https://journals.ametsoc.org/view/journals/amsm/58/1/amsmonographs-d-16-0006.1.xml

Additional discussion was included in lines 178 through 183 of the revised manuscript.

**2.    I think Figure 9 is the most important result in this study, but they simply classify the particles into three categories: C-, Cl-, and Si-based.  I suggest they discuss their particle categories more. I assume they are carbonaceous, sea salt, and dust particles. With particle shape analysis and more detailed compositional analyses, it would be possible to identify them. At least for simple classifications such as those in Fig. 9, there is no need to discuss SEM-EDS conditions, but a rough qualitative measurement can classify them.**

We acknowledge the importance of Figure 9 in our study and appreciate the suggestion to provide more detailed discussions on particle categories. In categorizing unknown particles within hailstones, we employed k-means clustering and silhouette scores for statistical clustering to reveal particle similarities (Demšar et al., 2013) (1). Subsequently, we identified three main

clustered groups based on the predominant elements C, Si, and Cl. These elements were determined by analyzing each particle cluster and determining the dominant element in each cluster.

The reviewer's assumption aligns with our findings, as the predominant elements (C, Si, and Cl) generally correspond to carbonaceous, dust, and salt particles, respectively. A similar correlation was previously observed by Lata et al. in 2021 (2). Their paper inspired our approach to create a decision flow chart to set elemental percentage thresholds to separate the particles into the categories found through the k-means clustering approach. However, it's important to note that carbonaceous particles may also originate from sources like soil dust, referred to as soil organic carbons, which are particularly common in arid and semi-arid areas (3). Furthermore, our analysis found instances where both silicates and carbonaceous particles exhibited the presence of heavier metals.  Therefore, we further distinguished the clusters dominated by C that contained heavy metals with those that did not, as well as Si-based particles also containing heavy metals. These additional categories are now also included in this revised manuscript.

We've included an illustrative example in the figure below to show a representative particle from each of the three major identified categories, allowing for more precise differentiation under CLSM and SEM. Hu et al. (2022) (4) is an example of a study that used the SEM to examine dust particles. By comparing the morphology of the dust particles in their study, for example, with images obtained through our study's method, we can draw comparisons with other particles previously studied using SEM. To emphasize the analytical capabilities of the methods employed in our study, including studying dust morphology, we will include the figure below in the revised text showing the confocal topography and SEM images for our different particle types.

[Figure]

Figure 11: Examples from each of the primary particle categories are presented, along with their ProFilmOnline topographical output obtained through a Confocal Laser Scanning Microscope (CLSM, top), accompanied by their corresponding Scanning Electron Microscopy (SEM, bottom) image.

(1) https://orangedatamining.com/
(2) https://pubs.acs.org/doi/10.1021/acsearthspacechem.1c00315
(3) https://www.mdpi.com/2073-445X/11/2/176
(4) https://www.sciencedirect.com/science/article/pii/S0048969722024081

Discussion and the figure have been added to the revised text from lines 287 through 304.

**3.   In the later discussion, there is very limited discussion of the results from the CLSM. I suggest more discussion using the results.**

We acknowledge the need for a more comprehensive discussion regarding the results obtained from the CLSM. We emphasized how we used it to locate the particle and ensure it was the same particle being analyzed in the SEM. However, CLSM also provides topographical and shape information about the analyzed particles, as seen in the figure in our previous response (which is included in the updated version of the manuscript). While a detailed analysis of the topography and shape of the particles is for a future publication focusing on additional applications of this method, this information does have implications for understanding ice nucleation processes in the cloud.

Surface topography is an active site for ice nucleation, influencing nucleation modes and the energy barrier for ice formation. This aligns with findings from Holden et al. in 2021 (1), suggesting that surface topography plays a significant role in nucleation. Additionally, other laboratory studies (2) have demonstrated that particle shape, size, and coating can impact the ice nucleation ability of particles, including soot. This paper shows that coatings or internal mixing have resulted in different ice nucleation abilities compared to bare particles. Importantly, these studies emphasize that particle size is not the sole determinant, highlighting the critical role of surface topography and particle shape in influencing atmospheric ice nucleation.

In our manuscript revision, we have incorporated a more thorough discussion of the CLSM results within the context of the figure discussed in the previous answer.

(1) https://www.pnas.org/doi/full/10.1073/pnas.2022859118
(2) https://acp.copernicus.org/articles/22/5331/2022/

**The discussion has been added in lines 277 through 286 of the revised manuscript.**

**4.   Although the authors have many discussions about the influence of glass plates, I suggest using a metal plate or a substrate consisting of single elements (e.g., Cu, Al, and**

**C). A glass plate contains so many elements that it interferes with the particle composition. By using other substrates, you will reduce such interferences.**

We appreciate your comment. Initially, we considered using a nickel base, as we anticipated finding elements in the hailstone, such as Al and C, based on previous studies (e.g., Michaud et al. 2014). However, using a glass base, we can examine the sample using transmitted light microscopes (i.e., CLSM), which would not be possible with a metal substrate, as glass is the only substrate transparent to visible light. Additionally, we encountered challenges with adhering hailstones to a metal surface, as we have yet to experiment to assess the adherence of ice to the nickel or other metal bases and the potential reactions of formvar during the sublimation process.

Given our need for using the CLSM, plus our exploration of various methods to prevent contamination, including methods to remove/subtract such contamination from spectral results, we have found that the current approach using glass is the most suitable for extracting the information we seek from the hailstones.

Additional text has been added to the revised manuscript to further support the use of glass as the substrate in lines 156 through 157.

**5. Did formvar coatings also contribute to the C signal? If so, C-based particles may need to be reconsidered.**

Good question. We compared potential C signals in clear glass sections (i.e., not containing visible particles) with and without formvar. Additionally, we examined C signals associated with a particle categorized as a carbonaceous to discern any significant differences in C presence. Our findings confirm minor C contributions from the formvar, as illustrated in the figure below. Due to its minimal thickness, the formvar layer's impact on the C signal is deemed negligible. We value this observation and have incorporated this figure and accompanying discussion into the revised text.

[Figure]

Figure 15: The left figure displays a single-point spectral analysis of a C-based particle with formvar (top panel), a clear area with formvar (middle panel), and a clear area without formvar (bottom panel). Each panel in the figure to the right is the corresponding panel providing a closer view of the C peak. Due to the formvar's thickness and the results in this figure, the formvar layer's impact on the C peak is considered negligible.

Additional text that includes this discussion was added in lines 345 through 350 of the revised manuscript.

**6. In the conclusion section, I do not think it is good to introduce other techniques such as Raman spectroscopy and STXM. They could be placed in the recommendation or other sections.**

Thank you for your recommendation. I agree with the suggestion, which has been addressed in the revised version of this paper.

The text was moved to lines 351 through 367 in the revised manuscript.

**7.   Could biological particles be identified by SEM using their specific morphology and composition, such as a tracer of P, S, and Cl?**

The SEM is a valuable tool for identifying biological particles based on their unique morphology and elemental composition. This imaging technique provides high-resolution, three-dimensional images of specimen surfaces, enabling the visual characterization of known biological structures such as cells, bacteria, and viruses. This can be coupled with EDS to analyze elemental tracers like phosphorus, sulfur, and chlorine.

However, a known biological particle must be used as a control for accurate comparisons with potential traces found in unknown particles within a hailstone. Our hailstone analysis method was initially developed using a hailstone sample that exhibited relatively weak traces of phosphorus and sulfur, rendering these elements unsuitable as references for distinguishing biological particles.

EDS, in general, cannot distinguish organic from inorganic carbon; this method encourages the exploration of supplementary methods (e.g., Kirchstetter et al., 2004 (1), Moffet et al., 2011 (2), Orlando et al., 2021 (3)) to enhance the overall understanding of biological particles, especially in scenarios involving subtle traces of specific elements.

(1) https://agupubs.onlinelibrary.wiley.com/doi/full/10.1029/2004jd004999
(2) https://digital.library.unt.edu/ark:/67531/metadc831481/
(3) https://doi.org/10.3390/chemosensors9090262

No additional text was added for this response.

**1.   Line 80: "Energy dispersive spectroscopy" should read "Energy dispersive X-ray spectroscopy".**

Thank you for this comment. This has been addressed in the revised manuscript.

This has been addressed in lines 79 and 80 of the revised manuscript.

**2.   Line 83: SEM works by scanning a focused beam of electrons.**

To address this point, we changed the text from "SEM works by focusing a beam of electrons onto the sample, which causes the emission of secondary electrons and backscattered electrons." to "SEM works by scanning a focused beam of electrons onto a sample, inducing the emission of secondary electrons and backscattered electrons."

This has been addressed in lines 82 through 83 of the revised manuscript.

**3.   Line 108: I think a sublimation point depends on both humidity and temperature, whereas a melting point depends only on temperature.**

Thank you for your comment. Both sublimation and the melting point of water depend on temperature and partial vapor pressure (which is proportional to humidity), as shown in the following figure. For this reason, we consider that no change is needed to the text.

[Figure]

No additional text was added for this response.

**4.    Line 170: I was not sure if sublimation in dry air can occur with silica gel. Although it can be determined by a detailed calculation, I think silica gel may not achieve the low humidity needed for sublimation. I am not sure about the current conditions, but it is better to check.**

We argue that the silica gel can achieve the low humidities required for sublimation because the hailstone sample from Central Argentina used in this case successfully underwent sublimation within a 48-hour timeframe. We are unsure what "current conditions" means and are open to discussing this point further.

No additional text was added for this response.

**5.    Line 197-200: Although a high vacuum SEM has a better SEM image than a low vacuum SEM, a low vacuum SEM has sufficient EDS capability for the purpose used in this study.**

Thank you for highlighting this aspect. Not all SEM models incorporate low vacuum environment capability, and during the development of our method, the SEM at our disposal lacked this feature.

Many SEMs with this capability to analyze samples with low vacuum conditions introduce a known gas, like nitrogen, or allow some oxygen from the environment to enter the chamber. However, this will raise potential concerns. One concern is contamination from the surrounding air. Without a clean oxygen source entering the chamber or a filter for purification (because SEMs do not have built-in air filtration systems), contaminants from the surrounding atmosphere could compromise the chamber. The hailstone consists of ultra-pure water, so it is susceptible to absorbing impurities in the air, which could impact the sample's integrity.

Another consideration under low vacuum conditions is atmospheric skirting, wherein a gaseous environment modifies the primary electron beam profile. The electron beam is typically divided into two fractions: an un-scattered beam with the original distribution profile and diameter and a scattered beam forming a "beam skirting" around it (1,2,3). This beam alteration occurs before reaching the particle surface, impacting the resolution of high-resolution imagery and spectral analysis through the EDS.

Therefore, due to potential issues related to contamination, lack of control over the beam's trajectory, and the resulting impact on resolution and spectral analysis, we respectfully disagree with the notion that low vacuum conditions would suffice for our method. We appreciate this observation; this discussion has been included in the revised version.

(1) https://www.sciencedirect.com/science/article/pii/S0065253908609026
(2) https://onlinelibrary.wiley.com/doi/abs/10.1002/sca.4950230505
(3) https://onlinelibrary.wiley.com/doi/abs/10.1002/sca.4950220304

Additional discussion was included in lines 204 through 217 of the revised manuscript.

**6.   Line 204: Is "sigma" OK?**

Thank you for this observation; it has been updated in the revised version to "Sigma."

The word has been corrected in line 222.

**7.   Line 208-209: A high voltage does not always improve the spatial resolution of SEM images due to its expansion in the materials. Please check again.**

Thank you for your comment. While the statement in question is supported by literature (Goldstein et al., 2017) (1), I agree with the reviewer that there are instances where excessively high voltages may lead to the expansion or penetration of electrons into the materials, potentially affecting the resolution negatively. We, therefore rewrote the sentence in the following way:

"The accelerating voltage of the primary beam determines the wavelength of electrons, and higher voltages are generally advised for enhanced spatial resolution in electron imaging

(Goldstein et al., 2017). However, it's important to consider that this principle may not universally apply to all materials. In some cases, excessively high voltages could result in electron expansion or penetration into the materials, potentially diminishing resolution. Therefore, it is essential to determine the optimal voltage, potentially opting for a lower one, when analyzing specific samples to ensure optimal imaging resolution."

(1) https://link.springer.com/book/10.1007/978-1-4939-6676-9

Correction was replaced in lines 226 through 231.

**8. Line 214: I believe that a working distance does not affect the beam diameter. Please check it.**

The relationship between working distance and beam diameter in SEM systems is well-established in existing literature (1,2). The working distance, representing the distance between the final lens of the SEM column and the specimen, influences the beam diameter, which is the diameter of the electron beam at the specimen. The electron optics within the SEM column governs this relationship.

As detailed in literature reference (1), the increase in working distance results in a proportional increase in the beam diameter. The geometric spreading of the electron beam over a greater distance from the final lens to the specimen contributes to this phenomenon.

Moreover, studies on the design and fabrication parameters for optical systems, including SEM (2), emphasize the interdependence of working distance and beam diameter. Specifically, variations in the working distance directly impact the beam diameter, with an increase in working distance corresponding to a larger beam diameter. This reinforces our statement in the paper, aligning with established literature on SEM systems.

(1)      https://www.sciencedirect.com/topics/engineering/lateral-resolution
(2)      https://www.mdpi.com/1424-8220/18/12/4150

No additional text was added for this response.

**9. Line 225: A 15 kV can measure higher than Fe. Please check it.**

The sentence reads: "The choice of 15 kV ensures that heavier elements are included in the EDS analysis by exciting the K lines of elements up to Fe." However, the intention was not to imply that we can measure up to Fe at a 15 kV accelerating voltage; instead, the emphasis is on the capability to measure heavier elements such as Fe at this voltage. It has been addressed in the revised version.

Correction was done in lines 244 through 245.

**10.  Line 237: There is no Figure 7D.**

Thank you for pointing out this discrepancy. The reference has been corrected to "7-B," aligning with the intended figure for citation in this text.

Text was replaced in line 257.

**11.  Line 239: I agree. Please see my major comment**

Thank you for your feedback. We acknowledge your agreement and have considered your major comment.

**12.  Line 258-260: If the particles have been classified according to these criteria, there is no need to use the cluster analysis (line 253-257).**

Thank you for the opportunity to clarify our classification approach. Going into the study, we did not know what types of particles we would find in our hailstone, so we started with a cluster analysis to see which elemental clustering was dominant in our sample. Utilizing k-means and silhouette scores analysis allowed us to identify particle similarities based on statistical clustering. Subsequently, with the knowledge of each particle's cluster, we determined the predominant element within each category (i.e., carbonaceous, silicates, salts.). Applying thresholds from the data, particles included in the C-based group had a C abundance greater than 10% weight, with this abundance being higher than that of Cl and Si. Those categorized in the C-heavy group met the same criteria as the C-based group but also had an abundance greater than 1% weight of heavier metals such as Ti, Cr, Fe, Ni, Zn, Br, and Mo. Particles categorized in the Si-based group had an SI abundance greater than 10% weight, with this abundance being higher than that of C and Cl. The Si-heavy group met the same criteria but had an abundance greater than 1% weight of heavier metals such as Ti, Cr, Fe, Ni, Zn, Br, and Mo. Finally, particles with a Cl abundance greater than 10% weight, with this abundance being higher than that of C or Si, were categorized in the Cl-based group.

Determining how the data should be categorized without this initial clustering would be challenging. Therefore, cluster analysis was necessary to ensure an appropriate classification of particles. We have emphasized this point in the revised manuscript.

Discussion and the figure have been added to the text from lines 287 through 304.

**13.  Line 275: Is it true that CCSEM can load samples automatically?**

Based on the information provided by SEM manufacturers, computer-controlled scanning electron microscopes offer automated sample-loading capabilities (1,2,3,4,5). For instance, the ZEISS EVO scanning electron microscope incorporates Automated Intelligent Imaging, a feature that enhances sample throughput. Additionally, it provides tools for relocating regions of interest and ensuring the integrity of collected data, thus facilitating automated and efficient

sample handling. Similarly, the JEOL FE-SEM has an AI system called NeoEngine, which tracks electron beam trajectories, streamlining operations with minimal user intervention. We added a reference to the revised paper to describe this capability.

(1) https://www.zeiss.com/microscopy/en/products/sem-fib-sem/sem/evo.html

(2) https://www.azom.com/article.aspx?ArticleID=19595

(3)https://www.thermofisher.com/us/en/home/electron-microscopy/products/desktop-scanning-electron-microscopes.html

(4)https://www.semtechsolutions.com/blog/better-performance-from-scanning-electron-microscope-use/

(5) https://www.hitachi-hightech.com/global/en/sinews/technical_explanation/130301/

No additional text was added for this response.

**14.    Line 276-277. I do not believe that CCSEM can measure thousands of particle compositions in less than an hour. EDS needs an acquisition time of at least several seconds.**

After reviewing your comment and cross-referencing it with my notes, I agree with your observation. More precisely, when we focused on analyzing individual particles, as described in this methods paper, we assigned a 2-minute acquisition time. Considering this time constraint, our analysis permitted the theoretical examination of approximately 30 single-point analyses per hour. These analyses could be for a single particle or encompass multiple spots within a single particle. To accommodate your observation, the sentence has been rewritten: "A CCSEM-EDS software can be programmed to autonomously analyze multiple particles consecutively, without requiring human intervention (Vander Wood, 1994)."

The text was replaced in lines 327 through 328.

**15.    Line 280: There is no section 2.4.2.**

Thank you for identifying this inconsistency; this has been updated in the revised version to "section 3.3.2."

The text was replaced in lines 330 and 332.

**16.    Line 285-288. I do not think changing the acceleration voltage is effective. First, when using low voltage first, you will not see heavy elements. Second, it is very time-consuming, as suggested in line 278.**

Adjusting the voltage, although time-consuming, becomes especially relevant when analyzing particles that are 1 micron or smaller in size. This is because the SEM has a higher resolution compared to the CLSM. This higher resolution in the SEM gives us a distinct advantage,

allowing for a more detailed examination of smaller particles. This prevents any elemental contribution from the base due to a large activation volume, as illustrated in Figure 8 and discussed in depth surrounding this figure.

No additional text was added for this response.

**17.    Figure 7. There is a missing peak identification around 2.1 keV. Why is this?**

The spectral peak corresponds to gold, as it was used in coating the sample and is consequently excluded from the analysis. I have incorporated this clarification into the figure caption.

Additional text was added to Figure 7's caption.

**18.    Figure 9. I cannot see the right images. Are they from Figure 6?**

Thank you for identifying this inconsistency; this has been updated in the revised version to "... Figure 5."

This inconsistency was addressed in Figure 9.

---

## Author Comment (AC4)

The manuscript, "Revolutionizing Hailstone Analysis: Exploring Non-Soluble Particles through Innovative Confocal Laser and Scanning Electron Microscopy Techniques," is an account of investigating hailstones by applying multiple analytical techniques and employing a special FORMVAR coating procedure to preserve the spatial distribution of particles captured within hailstone thin sections. While the manuscript makes several interesting points it suffers from some significant shortcomings. The most concerning of these in my opinion is that the manuscript really focuses on findings from what appears to be one thin section of one hailstone. It is very unclear how representative the results are and moreover, if the focus is more to present the methodology, they to not inspire belief that these types of experiments would be easy and straightforward to reproduce in a manner that would lead to statistically significant data.

Thank you for your review of our work. We understand the initial concern about showing results from one hailstone. We intended to include results comparing two hailstones for a follow-on paper, focusing on methodology. Still, we also recognize the need here to show confidence in the reproducibility of this technique. With the remaining laboratory hours available during this funding cycle, we completed an additional analysis to address your points below. Responses to your suggestions for improvement are provided below, including new figures from the analysis of new stones/stone sections.

Several areas for improvement are:

1. The hailstone preparation and sublimation method could benefit from a better descriptive illustration/figure. The utilization of FORMVAR seems to be a legacy technique that it is not common so that readers might have intuition about how it works.

Thank you for your observation. We do not have additional photos taken during the preparation phase, but we acknowledge the need to provide more detail on this technique. To this end, the following description has been added to the text in lines 171 through 175:

"After taking pictures of the hailstone sample after being polished, a layer of FORMVAR solution is applied to the sample's surface using a clean glass rod. This application is done in two ways: 1) by dipping one side of the rod into the FORMVAR solution and spreading a small amount over the surface, or 2) by pouring small amounts of the solution onto the surface and evenly spreading it across the polished hailstone. Once the entire surface is covered with the FORMVAR solution, the sample is ready for sublimation."

2. Links between particles and nucleating particles are quite tenuous. There are many particles in the analyzed sample and it appears impossible to deconvolute what was there when the ice began to form, versus what was accumulated during transport in the cloud etc.

Thank you for your comment. We acknowledge that distinguishing between particles present during the initial formation of the hailstone and those accumulated during transport is challenging. However, our approach to distinguishing which particles existed within the

hailstone's core or embryo is meant to highlight which particles were present where nucleation occurred and, thus, were likely involved in the initial nucleation process. These particles are distinguished from those in the outer layers more prone to being acquired during transport. A comparison of particles in the embryo versus outer layers will be shown in a follow-up paper and linked to likely sources in the region for separate hail events. Previous work in melting stones had isolated the embryo (e.g., 3 hailstones melted and analyzed in Michaud et al. 2014) to also make inferences on the composition of particles that have the potential to act as nucleation sites owing to their presence in the embryo. By avoiding the melting stage, we also preserve the in situ location of the particles with respect to the embryo and their sizes and shapes, thus advancing our knowledge of the types and characteristics of particles likely leading to ice nucleation in the formation of hailstones.

3. Figure 9 appears to be the most interesting result, but is difficult to interpret and the photographs that are included are extremely small.

Thank you for this observation. After reviewing Figure 9 and the information we aim to present, as well as taking into account your other suggestions to analyze a different cross-section of the same hailstone as well as other hailstones, we have decided to remake the figure and distribute the information across several new figures as follows:

Our new Figure 9 more clearly shows the two cross-section areas chosen to analyze particle size distribution and composition.

[Figure]

Figure 9: An example analyzed hailstone (V-7) where the areas highlighted by red rectangles indicate where particles were randomly selected to measure particle size distribution using CLSM and elemental composition via SEM-EDS. The orange circle marks the location of the embryo.

A new Figure 10 simplifies the message about the particle size distributions in comparing them among the two cross-sections in Figure 9-A,B (V-7V, V-7H) and two additional stones collected from the same storm (Figure 9-C,D; V-16, V-17). Note that the different axis ranges in this figure represent the different number of particles analyzed and differences in size ranges owing to the different sizes of the hailstones and, thus, cross sections. The results of this new figure are described below in response to your related suggestion to analyze additional stones.

[Figure]

Figure 10: Particle size distributions for hailstone samples collected during the event on 8 February 2018. Panels a) and b) display particle sizes for two different cross-sections from sample V-7, while panels c) and d) show particle sizes for samples V-16 and V-17, respectively.

In our original Figure 9, the details of the composition of particles were obscured by overlapping dots; thus, we have added a new Figure 12 to the manuscript. This figure shows the elemental composition distribution of particles for V-7V (Figure 12-A) and V-7H (Figure 12-) cross-sections (i.e., different cross-sections within the same stone). This figure more clearly demonstrates the dominant elemental compositions within the stone. It shows

that the relative contributions of different compositions are similar when analyzing two different cross-sections of the same sample (i.e., V-7V, V-7V).

[Figure]

Figure 12: EDS-based elemental composition distribution of particles for a) V-7V and b) V-7H cross-sections, as seen in Figure 9.

Finally, we would like to highlight a benefit of this method in that we can isolate particles within the embryo compared to the outer layers, and in our case, as previously highlighted, being able to describe both the sizes and composition of the particles that may have served in the nucleation process of this stone. Our new Figure 13 isolates just those particles in the embryo regions of the cross sections. Both V-7V (Figure 13-A) and V-7H (Figure 13-B) cross through the embryo (see new Figure 9). Still, different particles were selected within the embryo sample to elucidate better the range of particle characteristics observed within this stone's embryo. Also, because the lead author has gotten more efficient at this process, he was able to analyze more particles in a similar amount of time, thus the difference in particle numbers in the new V-7H analysis. The messages from these figures are described below in response to another of your suggestions.

[Figure]

Figure 13: EDS-based elemental composition distribution of particles found in the embryo for sample V-7 for a) V-7V and b) V-7H.

With these new figures, we address your remaining comments below and have enhanced the interpretability of the information presented in the results section of the revised manuscript.

4. Given the lack of duplicates etc. (see comment above) it is very hard to assess the utility of all of the effort that went into this analysis. If one were to take a second thin section of the same stone and repeat the analysis, would we get wildly different results, or similar? What do we learn in either case? What about with another stone from the same storm? Is it even practical to do this work on many stones?

Thank you for this suggestion. Thankfully, we had remaining lab hours budgeted to analyze the particles in a horizontal section through V-7 (V-7H as in new Figure 9). We analyzed particles within the same sectors covering the embryo as in our original map (i.e., V-7V as shown in new Figure 9), including CLSM-based size distributions and EDS-based elemental composition. Due to increased proficiency with this technique, we could analyze more particles in a shorter amount of time. The results of this second cross-section are shown in the revised manuscript as new Figures 9, 10, 12, and 13. Furthermore, we had two other hailstones from this same storm (V-16, V-17) that we could analyze in the size distribution from the CLSM for comparison with the V-7 stone, with results included in the new Figure 10.

Our analysis of this different section in the same hailstone revealed that particles were overall smaller in the V-7H (Figure 10-A) cross-section than in V-7V (Figure 10-B) but still contained a few isolated particles exceeding 100 microns. Additionally, compared to V-16 (Figure 10-C) and V-17 (Figure 10-D), particle sizes in V-7, the smallest hailstone of the three, are relatively smaller (Figure 10). Regarding elemental composition (Figure 12-A,B), we did not find significant differences; carbonaceous particles remained predominant, with silicates being the second most dominant particles. In the embryo region (Figure 13), we discovered that while salts were not identified in the initial analysis of V-7V (Figure 13-A), they were present in the additional horizontal cross-section (Figure 13-B; V7-H), along with

heavier metals. These new figures and their interpretation are included in the Results section of the revised manuscript.

In summary, these figures demonstrate the robustness of our method, showing overall consistent messages but highlighting the value of looking at multiple cross sections in one stone, particularly for identifying a variety of elemental components. Across hail stones from the same storm, there appears to be an increase in particle sizes with increasing size of the hailstone that is an interesting result to explore further, showing the value of this unique method that preserves particles for analysis of both size and composition with respect to the embryo. Although we only show the size and composition within the embryo region here, we have extended this analysis to compare with the outer layers and in comparing results with a hailstone from a different storm under different environmental conditions, and therefore, regional transport sources that are the subject of a soon-to-be submitted paper further highlighting results of this unique method.

5. The CLSM work that lays the foundation for SEM analysis appears to resolve particles down to 1 micron.  This is still quite a large size, and many particles will be much smaller than this. Even ice nucleation parameterizations are largely based on particles with sizes greater than 0.5 microns. Many such particles missed here.

We acknowledge that this CLSM analysis will result in missing particles smaller than 1 micron, including those down to 0.05 microns that may serve as INPs. However, the flexibility of our method allows for the analysis to begin with SEM, which provides higher resolution and ensures these smaller particles are not overlooked. Although SEM cannot provide topographical information of the particles, starting with SEM enables the detection of particles smaller than 1 micron. We started with CLSM to get more detail about the individual particles' size/shape/topography. Also, a limitation of analyzing smaller particles with the SEM is that this method requires a glass substrate, which may introduce spectral contamination at smaller sizes. These limitations are included in the revised manuscript between lines 242 through 252.

My overall reaction to the submitted manuscript is that in its current form the work falls short of a new atmospheric measurement technique, or some protocol that could be widely adopted. Rather it is a report on the application of several analytical methods to a single hailstone (or single thin section from a single hailstone) from a unique event.  The authors mix cases and do at times refer to the plural "hailstones".  If they have more data, I would encourage them to complement the manuscript to find more general conclusions.  Without this I do not see the extension to the interests of a general readership. That said, I do complement the authors on the incorporation of citizen science.

Thank you for your comments and the opportunity to address your concerns.

While we primarily discuss a singular hailstone in this manuscript to describe the method proposed for hailstones, we have added more information from another section of the same hailstone and results from two other hailstones from the same storm. Our intention here in this paper is to describe the methodology in detail, its adaptability, limitations, and potential for broader applications, with more results from hailstones in multiple events being the focus of a separate detailed results paper given the length of this detailed methodology.

In response to your observations:

1. We have improved our efficiency with the technique, enabling us to analyze twice as many samples as before in the same amount of time. This increased proficiency demonstrates the potential for scalability and broader application of our methodology.

2. We acknowledge the limitations of the current dataset due to time constraints. However, the lead author is committed to continuing this work post-PhD, further refining the methodology and expanding the dataset to draw more general conclusions for hailstones within and outside Argentina (i.e., in the U.S.).

3. We believe that with practice and incorporating lessons learned as detailed in this methodology-focused manuscript, others can replicate and build upon our methodology. We are open to sharing our experiences and providing guidance to facilitate other researchers' adoption of this technique following the publication of this paper.

We appreciate your recognition of our incorporation of citizen science and will strive to continue engaging with the broader public and scientific community.